# Systematic screening of generic drugs for progressive multiple sclerosis identifies clomipramine as a promising therapeutic

Simon Faissner[1,2], Manoj Mishra[1], Deepak K. Kaushik[1], Jianxiong Wang[1], Yan Fan[1], Claudia Silva[1], Gail Rauw[3], Luanne Metz[1], Marcus Koch[1] & V.Wee Yong[1]

The treatment of progressive multiple sclerosis (MS) is unsatisfactory. One reason is that the drivers of disease, which include iron-mediated neurotoxicity, lymphocyte activity, and oxidative stress, are not simultaneously targeted. Here we present a systematic screen to identify generic, orally available medications that target features of progressive MS. Of 249 medications that cross the blood–brain barrier, 35 prevent iron-mediated neurotoxicity in culture. Of these, several antipsychotics and antidepressants strongly reduce T-cell proliferation and oxidative stress. We focus on the antidepressant clomipramine and found that it additionally inhibits B-lymphocyte activity. In mice with experimental autoimmune encephalomyelitis, a model of MS, clomipramine ameliorates clinical signs of acute and chronic phases. Histologically, clomipramine reduces inflammation and microglial activation, and preserves axonal integrity. In summary, we present a systematic approach to identify generic medications for progressive multiple sclerosis with the potential to advance rapidly into clinical trials, and we highlight clomipramine for further development.

[1] Department of Clinical Neurosciences, Hotchkiss Brain Institute, University of Calgary, Calgary, AB T2N 4N1, Canada. [2] Department of Neurology, St. Josef-Hospital, Ruhr-University Bochum, 44791 Bochum, Germany. [3] Neurochemical Research Unit, Department of Psychiatry, University of Alberta, Edmonton, AB T6G 2B7, Canada. Correspondence and requests for materials should be addressed to V.W.Y. (email: vyong@ucalgary.ca)

Multiple sclerosis is a multifactorial inflammatory condition of the CNS leading to damage of the myelin sheath and axons/neurons followed by neurological symptoms[1]. Approximately 85% of multiple sclerosis patients present with a relapsing-remitting phenotype and the majority of these evolve to a secondary-progressive disease course after 15–20 years. 10–15% of the patients experience a primary progressive disease course with slow and continuous deterioration without definable relapses.

While there have been tremendous successes in the development of medications for relapsing-remitting multiple sclerosis during the last decade, nearly all studies conducted in progressive multiple sclerosis have failed such as the recently published INFORMS study on the sphingosine-1-phosphate inhibitor fingolimod[2]. However, the results of two Phase 3 trials of ocrelizumab in primary progressive[3] and siponimod in secondary progressive multiple sclerosis[4] were announced recently to have met their primary outcomes. Orcelizumab has since been approved for use in primary progressive MS.

The reasons for the general lack of medications in progressive multiple sclerosis are manifold. One explanation is that the underlying pathology of progressive multiple sclerosis has profound differences to the relapsing-remitting form[5]. Examples include the more pronounced neurodegenerative aspects of the progressive disease in conjunction with significant mitochondrial damage[6,7], iron accumulation which contributes to the elevated oxidative stress from several sources[8,9], and the more common representation of B-cell follicular structures underneath the meninges in progressive cases[10,11]. Moreover, the blood–brain barrier in progressive multiple sclerosis appears to be repaired compared to the breach in relapsing-remitting disease[5], so medications will require the capacity to enter the CNS.

Another pathologic feature seen in all types of multiple sclerosis but appears exacerbated in progressive cases is intense focal microglia activation[5,12,13]. We previously conducted a systematic screen on microglia inhibition using the drug library of the NINDS Custom collection II (US Drug Collection); the majority of compounds in this library are generic medications. Out of 1040 compounds, 123 reduced tumor necrosis factor alpha (TNF-α) production by activated microglia by over 50%[14]. Based on this research, we expanded our screen to investigate other features relevant to progressive multiple sclerosis, including the potential of generic compounds to affect iron-mediated neurotoxicity, maintain mitochondria integrity, and scavenge free radicals. We sought also to shortlist a compound for further activity against T-lymphocytes and B-lymphocytes, given that the adaptive immune response continues to be active within the CNS compartment in progressive multiple sclerosis[15], and we sought to interrogate whether the compound affects experimental autoimmune encephalomyelitis (EAE), a model of MS. Out of 249 investigated medications, 35 prevented iron-mediated neurotoxicity in culture. Out of these, several reduced the proliferation of T-lymphocytes and had antioxidative potential. The tricyclic antidepressant clomipramine also affected B-lymphocyte proliferation, reduced clinical signs in acute EAE concomitant with improved histology, and improved the chronic phase in two EAE models.

selected drugs that are orally available, for ease of use, this does not imply that injectable medications would not be effective medications in progressive multiple sclerosis, as illustrated by ocrelizumab recently[3]. Out of the original list, 791 compounds were thus excluded and 249 were selected for further testing. The detailed information of each of the 249 compounds is provided in Supplementary Dataset 1.

The 249 compounds were first tested against iron toxicity to human neurons in culture. Neurons were pre-incubated with each compound for 1 h followed by application of $FeSO_4$. Ferrous iron (25 and 50 μM) is very toxic to neurons, with >80% loss of microtubule-associated protein-2 (MAP-2)-labeled neurons by 24 h in most experiments compared to the control condition (Supplementary Figs. 1–7, Supplementary Data set 2). An example of iron toxicity and a drug screen is shown in Fig. 1. Of all drugs tested, 35 compounds showed statistically significant protection from $FeSO_4$-mediated neurotoxicity (Fig. 2a). Of these, antipsychotics, such as clozapine or periciazine, and tricyclic antidepressants, such as clomipramine or desipramine, exhibited strong protection, as shown after normalization across at least 2–4 experiments (n of four wells of cells per experiment per test condition) to the number of neurons of the respective control conditions (Fig. 2a). For example, while the average loss of neurons over 24 h in response to $FeSO_4$ was 85.5% (i.e., 14.5% of surviving neurons compared to 100% of controls), clomipramine at 10 μM completely prevented neuronal loss (107.3% of controls). Other categories of medications with neuroprotective actions against iron included antihypertensives and some antibiotics. We note that minocycline, an antibiotic that reduces the conversion of a first demyelinating event to clinically definite multiple sclerosis in a Phase 3 clinical trial, was not included in the 1040 compounds; in a separate study, we find minocycline to completely prevent iron neurotoxicity as well[16].

Live-cell imaging over 12 h supported the neuroprotective effects of drugs. We selected indapamide and desipramine for live imaging studies. Figure 2b and Supplementary Video 1 show that while the number of neurons with intracellular propidium iodide (PI), a dye that leaks across a compromised plasma membrane, in response to $FeSO_4$ exposure increases progressively over 12 h, this was significantly attenuated by indapamide and desipramine.

The 35 hits were further narrowed concerning their ability to cross the blood–brain barrier according to drugbank.ca, their side effect profile and tolerability. Although antipsychotics are not well tolerated they were further included in the screening due to their good blood–brain barrier penetrance. Out of these, a group of 23 compounds was chosen for their ability to prevent mitochondrial damage using rotenone, which inhibits the electron transfer from complex I of the respiratory chain to ubiquinone. Rotenone induced strong neurotoxicity to neurons (Fig. 3). The tricyclic antidepressant trimipramine, the antipsychotics clozapine and periciazine, promethazine and the antihypertensives labetalol, methyldopa and indapamide reduced neurotoxicity, while clomipramine trended toward a protective activity (Fig. 3a). The effect size of rescue by medications was, however, small. Of note, rotenone induced marked morphological neuronal changes with retraction of neurites (Fig. 3c).

## Results

**Generics protect against iron and rotenone neurotoxicity.** Of the 1040 compounds available in the NINDS Custom Collection II, we first conducted a search of available information to exclude those that were either experimental, agricultural, not available as oral drug, not listed at Health Canada, steroid hormones or veterinary medications. Moreover, we omitted those that were not known to cross the blood–brain barrier. We note that while we

**Hydroxyl radical scavenging capacity of medications.** The biochemical cell-free hydroxyl radical antioxidant capacity (HORAC) assay investigates the prevention of hydroxyl radical-mediated oxidation of fluorescein in comparison to the strong antioxidant gallic acid. The generation of hydroxyl radicals by a cobalt-driven Fenton-like reaction oxidizes fluorescein with progressive loss of fluorescence. The presence of an antioxidant reduces the loss of fluorescence over time. As noted in Fig. 4a,

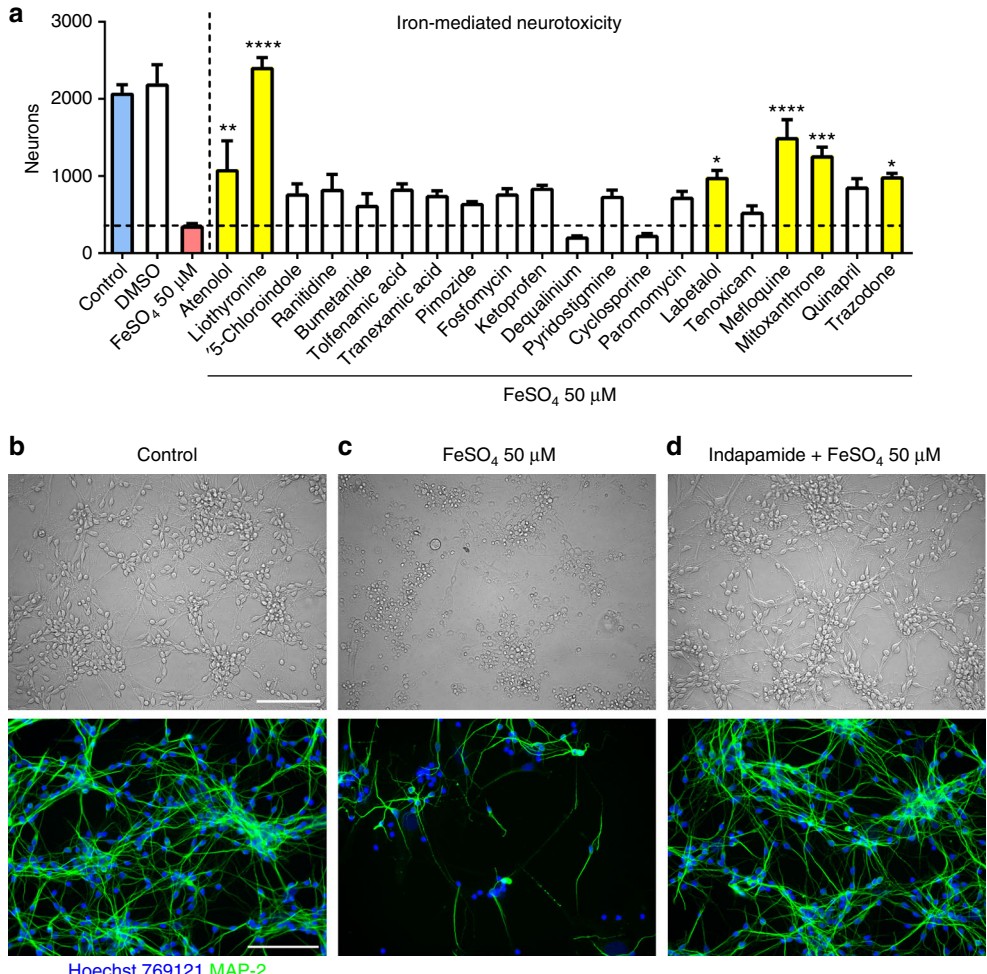

**Fig. 1** Screening of generic compounds to prevent iron-mediated neurotoxicity. Shown is an example of a screening of drugs to identify those that prevent iron-mediated neurotoxicity to human neurons. Neurons were pretreated with drugs at a concentration of 10 μM, followed by a challenge with 25 or 50 μM FeSO₄ after 1 h. In this experiment, several compounds (yellow bars) prevented against iron-mediated neurotoxicity as determined by the number of survival neurons in automated counts after 24 h (**a**). Values in **a** are mean ± SEM of $n = 4$ wells per condition. One-way analysis of variance (ANOVA) with Bonferroni post hoc analysis vs. iron: *$p < 0.05$; **$p < 0.01$; ***$p < 0.001$; ****$p < 0.0001$. **b–d** Representative images show the untreated control and iron-treated neurons, as well as the prevention of neurotoxicity by indapamide (top: bright field; bottom: fluorescence microscopy). Neurons were detected by anti-MAP-2 antibody. The scale bars depict 100 μm

gallic acid reduced the loss of fluorescence (upward shift) compared to a blank Fenton-driven reaction that is without antioxidant, while indapamide has an even higher activity.

We compared the area under the curve of test compounds to that elicited by gallic acid to obtain the gallic acid equivalent (GAE). A GAE of 1 represents hydroxyl radical scavenging capacity similar to that of gallic acid, while a compound without antioxidant activity would produce a GAE close to 0. Some of the compounds tested exhibited stronger antioxidative properties than gallic acid with HORAC-GAEs >1 (Fig. 4c). These included indapamide (mean HORAC-GAE 4.1; $p < 0.05$; one-way ANOVA with Dunnett's multiple comparisons test as post hoc analysis vs. gallic acid), mitoxantrone (5.6; $p < 0.001$), chlorpromazine (5.9; $p < 0.001$), clozapine (4.6; $p < 0.05$), and trimipramine (4.2; $p < 0.05$). Although not statistically significant compared to gallic acid, clomipramine had a HORAC-GAE of 2.1. Regarding the comparison to the blank condition (i.e., no antioxidant present), there was a significant upward shift by clomipramine of the slope over 60 min ($p < 0.0001$; two-way ANOVA with Dunnett's multiple comparisons test as post hoc analysis) (Fig. 4b). Thus, although clomipramine lacked significance against the strong antioxidative gallic acid, the compound exhibited strong

antioxidative effects against the blank situation (in the absence of any antioxidant). Interestingly, the tricyclic antidepressant desipramine had strong oxidative effects (HORAC-GAE −5.00; $p < 0.0001$).

**Proliferation of T-lymphocytes is reduced by antidepressants.** We tested the capacity of compounds to affect T-cell proliferation (Fig. 5). Splenocytes activated by anti-CD3/anti-CD28 to trigger the proliferation of T-cells had reduced incorporation of ³[H]-thymidine upon treatment with dipyridamole (mean reduction 89.3%; $p < 0.0001$; one-way ANOVA with Dunnett's multiple comparisons test as post hoc analysis compared to activated splenocytes), cefaclor (23%; $p < 0.01$), labetalol (26.8%, $p < 0.0001$ for this and subsequent compounds listed here), mefloquine (62.3%), mitoxantrone (99.7%), trimeprazine (43.3%), chlorpromazine (99.4%), periciazine (28%), promethazine (74.6%), clomipramine (68.2%), desipramine (92.2%), imipramine (66.4%), trimipramine (54%), and doxepin (85.3%, all $p < 0.0001$). Of note, methyldopa and memantine increased proliferation (methyldopa 41.4%, $p < 0.0001$; memantine 17.5%, $p < 0.05$). Mitoxantrone and chlorpromazine, however, had toxic effects (data not shown).

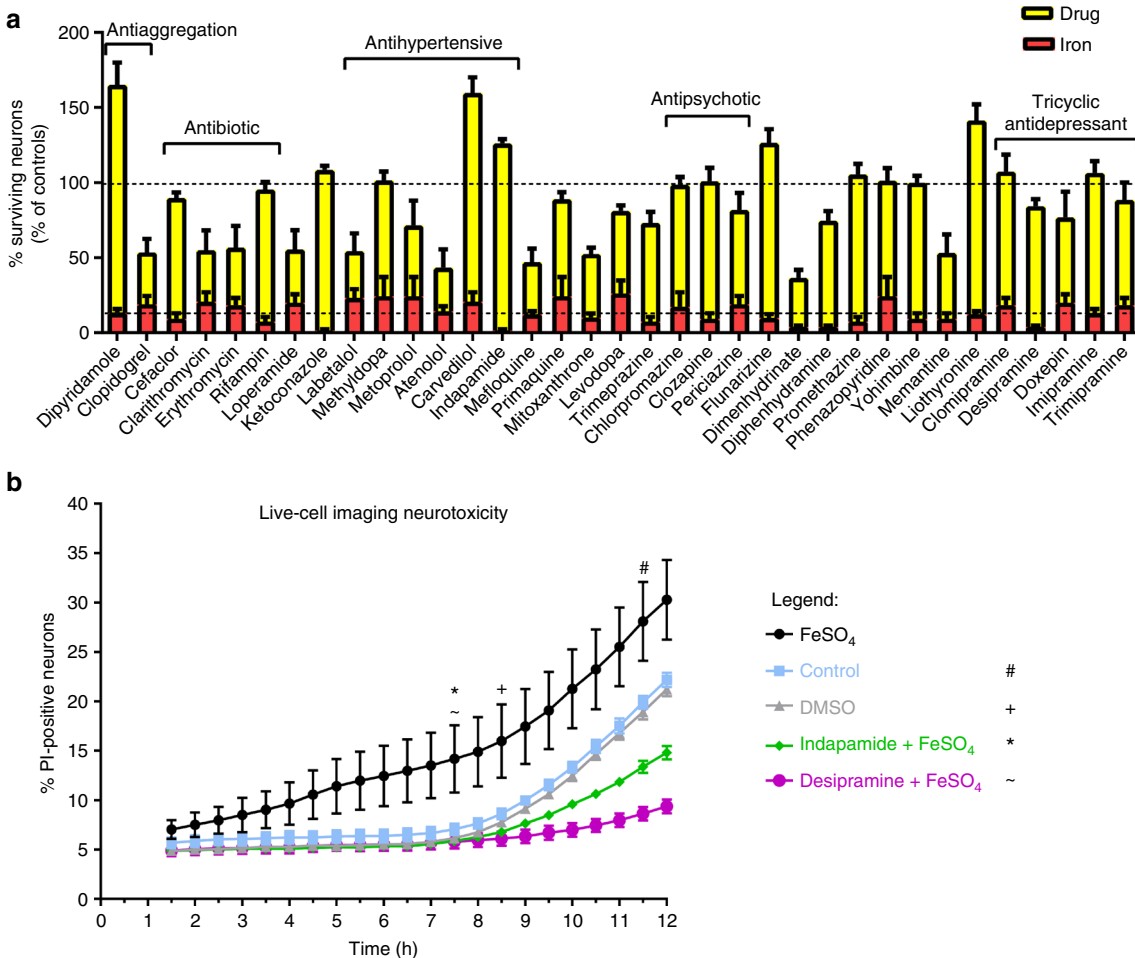

**Fig. 2** Summary of compounds that attenuate iron-mediated neurotoxicity. Shown are all 35 generic drugs that prevent iron-mediated neurotoxicity (**a**). The number of neurons in each well of a given experiment was normalized to the number of neurons of the respective untreated control condition (100%). The corresponding $FeSO_4$-treated condition (red) was also normalized to the respective control. Some of the major drug classes are depicted in the figure. Shown are the mean ± SEM of 2–4 independent experiments, performed in quadruplicates (thus, 8–16 wells per treatment across experiments are depicted in the figure). **b** shows the results from live-cell imaging of neurons (see examples in Supplementary Video 1), challenged with $FeSO_4$ in a concentration of 50 µM. Upon pre-treatment with indapamide or desipramine 1 h before the addition of iron, the number of propidium iodide-positive cells was significantly reduced after 7.5 h and even below the level of the untreated control condition after 12 h, suggesting a strong neuroprotective effect. Live-cell imaging was performed over 12 h, where images were taken every 30 min. The time-point from which significant changes from $FeSO_4$ were observed for each group is marked with a symbol (# control; + DMSO; * indapamide; ~ desipramine). Shown are means ± SEM of $n = 3$ wells per condition. Results were analyzed with a two-way ANOVA with Dunnett's multiple comparison as post hoc analysis

As indapamide did not reduce T-cell proliferation, we did not pursue it further in the T-cell prominent disease, EAE.

**Focus on clomipramine in vitro and in acute and chronic EAE**. We selected clomipramine for further study as it is a well-tolerated antidepressant and crosses the blood–brain barrier very well (drugbank.ca). Moreover, in our assays, clomipramine showed strong effects against iron-mediated neurotoxicity (mean % anti-MAP-2-positive cells normalized to control of 107.3%, representing complete protection against iron toxicity) (Fig. 2), had antioxidative properties (HORAC-GAE 2.1 where the effect of the antioxidant gallic acid is normalized at 1) (Fig. 4), and reduced T-lymphocyte proliferation (by 68.2%) (Fig. 5). We began with a concentration response with the intent of investigating lower concentrations since plasma concentration in human of clomipramine as an antidepressant average 122 ng/ml (387 nM)[17], but can peak to more than 600 nM in some individuals[18]. Figure 6a shows that clomipramine had a progressive significant increase in neuroprotection against iron toxicity from 100 nM.

The effect was mediated in part by chelation with iron, as washing away clomipramine from neurons led to cell death, while pre-incubation with iron before application to neurons totally preserved neuronal viability (Fig. 6b). We were able to observe the protection by clomipramine in a live-cell imaging study, in which the increasing number of PI-positive neurons over time in response to iron was attenuated by clomipramine (Fig. 6c, Supplementary Video 2).

T-lymphocyte proliferation was reduced in a concentration-dependent manner by clomipramine but significant reduction occurred only from 5 µM ($p < 0.01$; one-way ANOVA with Dunnett's multiple comparisons test as post hoc analysis compared to activated T-lymphocytes) (Fig. 6d). This was reflected by a cell cycle arrest with more cells in G1 ($p < 0.05$) and less in the S-phase ($p < 0.05$) from 2 µM (Fig. 6e, f).

Due to the growing knowledge about the importance of B-cell follicular structures for progressive multiple sclerosis[19,20], we sought to evaluate the effect of clomipramine on B-cell activation. BCR/anti-CD40L/IL-4 activation of B-cells increased their proliferation and production of TNF-α (Fig. 6g, h) and these

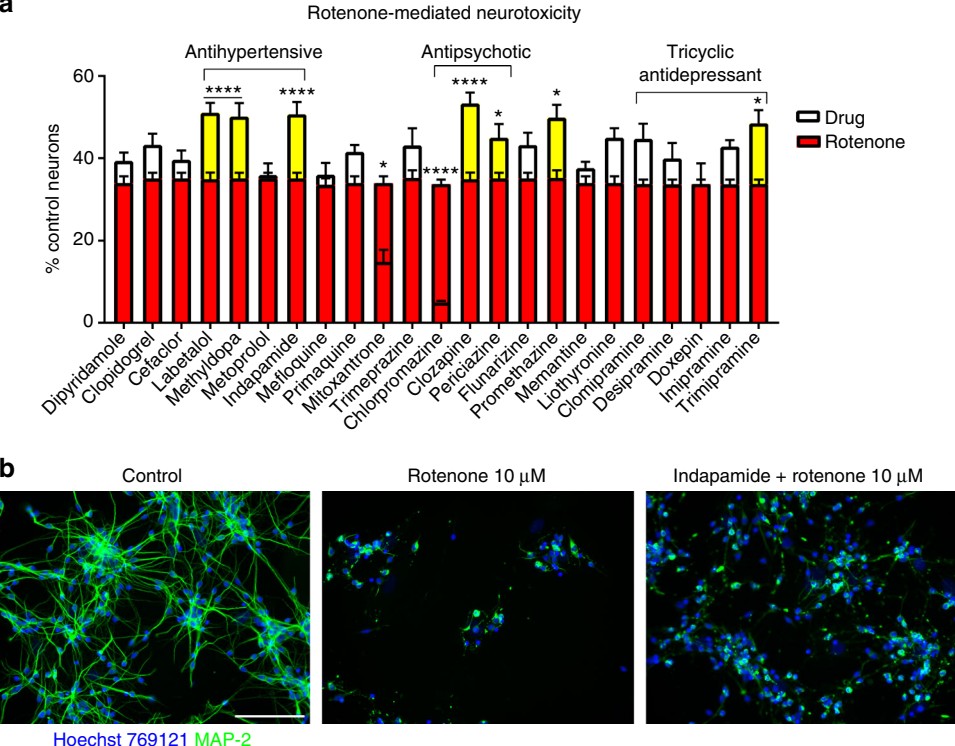

**Fig. 3** Prevention of mitochondrial damage induced by rotenone. Some of the generic drugs that prevented against iron-mediated neurotoxicity were tested against mitochondrial damage to neurons. Some compounds, such as indapamide, prevented mitochondrial damage as shown after normalization to the control neurons (**a**). However, the rescue effect was small. Treatment with rotenone induced marked morphological changes with retraction of cell processes (**b**). The scale bar shows 100 μM. Shown are normalized data of mean ± SEM of 1–3 experiments each performed in quadruplicates. Two-way ANOVA with Bonferroni multiple comparisons test as post hoc analysis vs. rotenone: *$p < 0.05$; ****$p < 0.0001$

were reduced in a concentration-dependent manner by clomipramine from 2 μM.

We then investigated clomipramine in acute EAE. Therapy with clomipramine from day 5 after induction of MOG-EAE delayed onset of clinical signs by 2 days with a significantly better early disease course between days 11 and 18 (Fig. 7a), which was reflected in an overall lower burden of disability (Fig. 7b). However, eventually, clomipramine-treated animals succumbed to EAE and increased disability (Fig. 7a).

We then sought to investigate whether initiation of treatment from the day of MOG-induction could improve the outcome of EAE. Remarkably, early treatment initiation completely suppressed the manifestation of clinical signs (Fig. 8a). While most animals in the vehicle group had a high disease burden, as shown by the sum of scores for each individual animal (Fig. 8b) and weight loss (Fig. 8c), this was profoundly ameliorated in treated mice over the course of study. PCR analyses of the spinal cord revealed that the significant elevation in vehicle-EAE mice of transcripts encoding *Ifng*, *Tnfa*, *Il-17*, and *Ccl2* were abrogated in clomipramine-EAE mice (Fig. 8d).

Investigation of serum levels of clomipramine and its active metabolite, desmethylclomipramine (DMCL), in mice sacrificed 1 h after the last of 16 daily clomipramine injections showed mean concentrations of 751 and 101 nM, respectively (Fig. 8e). Incredibly, the corresponding mean spinal cord levels were 28 and 1.5 μM; a similar high brain to plasma ratio of clomipramine was reported by Marty et al.[21] in mice injected with a single 8 mg/kg clomipramine IP. There was a strong correlation of serum and spinal cord levels for both clomipramine and DMCL across mice (Fig. 8f).

Histological analysis of the spinal cord showed profound parenchymal inflammation in vehicle-treated animals with a

histological score of 4.3, whereas clomipramine-treated animals only had few inflammatory cells in the meninges (score 1.7; $p < 0.001$; non-parametric two-tailed Mann–Whitney test) (Fig. 9a, b, g) that were inadequate to produce clinical manifestations as noted in Fig. 8a. Infiltration in vehicle-treated animals was accompanied by massive microglial activation, whereas clomipramine treatment prevented microglial activation, as assessed by Iba1 staining ($p < 0.01$) (Fig. 9c, d, h). Furthermore, clomipramine-treated animals had significantly less axonal damage ($p < 0.01$) (Fig. 9e, f, i). Infiltration and microglial activation correlated with axonal injury (Spearman $r = 0.7599$, $p < 0.01$; Spearman $r = 0.774$, $p < 0.01$, respectively; non-parametric two-tailed Spearman correlation with 95% confidence interval) (Fig. 9j, k).

We next set out to investigate the effect of clomipramine in chronic EAE. We first evaluated clomipramine initiated only after the first relapse when mice were in remission (day 31). In our hands, using the more sensitive 15-point EAE scoring system (rather than the conventional 5-point scale), MOG-EAE mice can be documented to undergo a second relapse after a remission period. Clomipramine did not affect the severity of the second relapse when initiated in mice at remission (Fig. 10a), likely because substantial neural injury had already occurred from a prolonged EAE course.

In another experiment, we treated MOG-immunized C57BL/6 mice from the first onset of clinical signs (day 13, Fig. 10b). Treatment with clomipramine attenuated the marked rise in clinical disability and had a significant positive effect during days 14–20 ($p = 0.0175$; non-parametric two-tailed Mann–Whitney test). During remission, likely because the severity of disability was low, the vehicle and clomipramine-treated groups did not differ. Disease was then followed by a second increase in clinical scores in vehicle-treated mice, which was prevented by clomipramine (days 42–50; $p = 0.0007$).

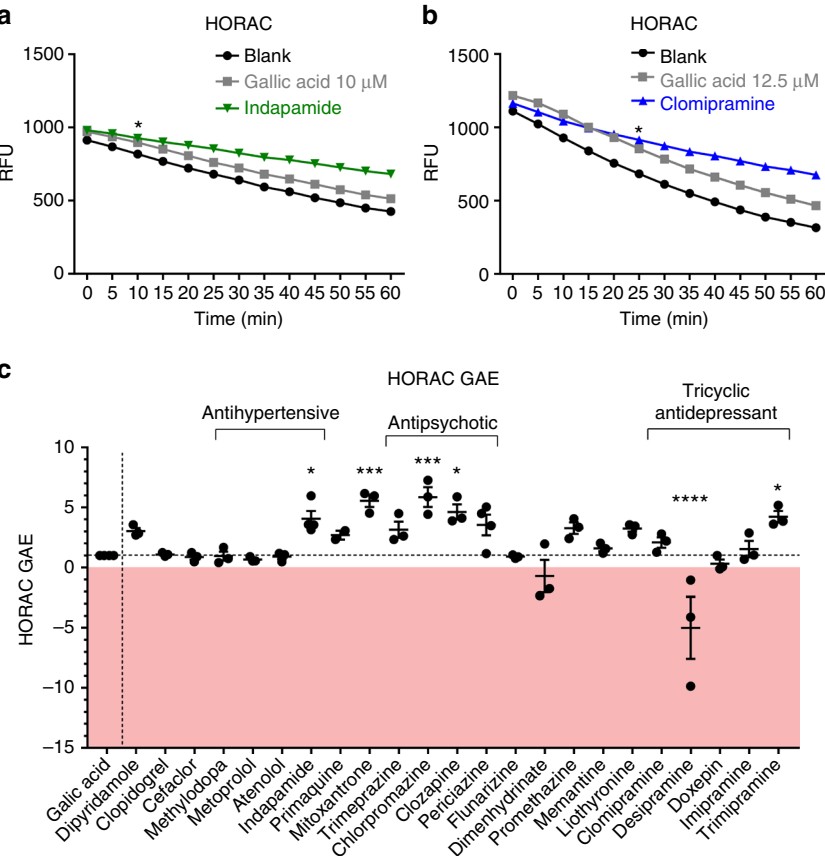

**Fig. 4** Scavenging of hydroxyl radicals in a biochemical assay. The antioxidative capacities of selected compounds that reduced iron-mediated neurotoxicity were analyzed using the HORAC assay. **a** shows a representative experiment depicting the decay of relative fluorescence units over 60 min for indapamide, gallic acid (GA) and the control (blank). **b** The upward shift of the curve for clomipramine in the HORAC assay indicates an antioxidative effect that is even stronger than gallic acid. HORAC gallic acid equivalents (GAEs) were calculated by the integration of the area under the curve of the decay of fluorescence of the test compound over 60 min in comparison to 12.5 μM gallic acid and blank. Shown are data of $n = 3–4$ independent experiments $\pm$SEM, with each experiment performed in triplicates (**c**). The antipsychotics showed strong antioxidative effects, as demonstrated with HORAC GAEs of >3. Data points >1 represent antioxidative capacity (the gallic acid effect is 1), 0 represents no antioxidative properties, and data <0 show pro-oxidative effect. Two-way ANOVA with Dunnett's multiple comparisons test as post hoc analysis (**a**, **b**); the first significant time point vs. gallic acid is depicted as asterisk. One-way ANOVA with Dunnett's multiple comparisons test as post hoc analysis vs. gallic acid (**c**). $*p < 0.05$; $**p < 0.01$; $***p < 0.001$; $****p < 0.0001$. RFU relative fluorescence units

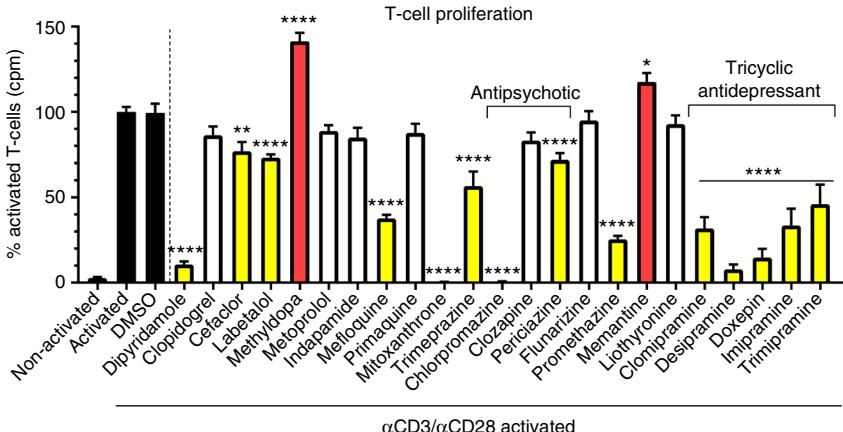

**Fig. 5** Effects on proliferation of T-lymphocytes. The tricyclic antidepressants (clomipramine, desipramine, imipramine, trimipramine, and doxepin) reduced proliferation of T-cells markedly ($p < 0.0001$). Data were normalized to counts per minute (cpm) of activated control T-cells. Shown are data pooled from two independent experiments each performed in quadruplicates. Data are depicted as mean $\pm$ SEM. One-way ANOVA with Dunnett's multiple comparisons test as post hoc analysis compared to activated splenocytes. $*p < 0.05$; $**p < 0.01$; $****p < 0.0001$

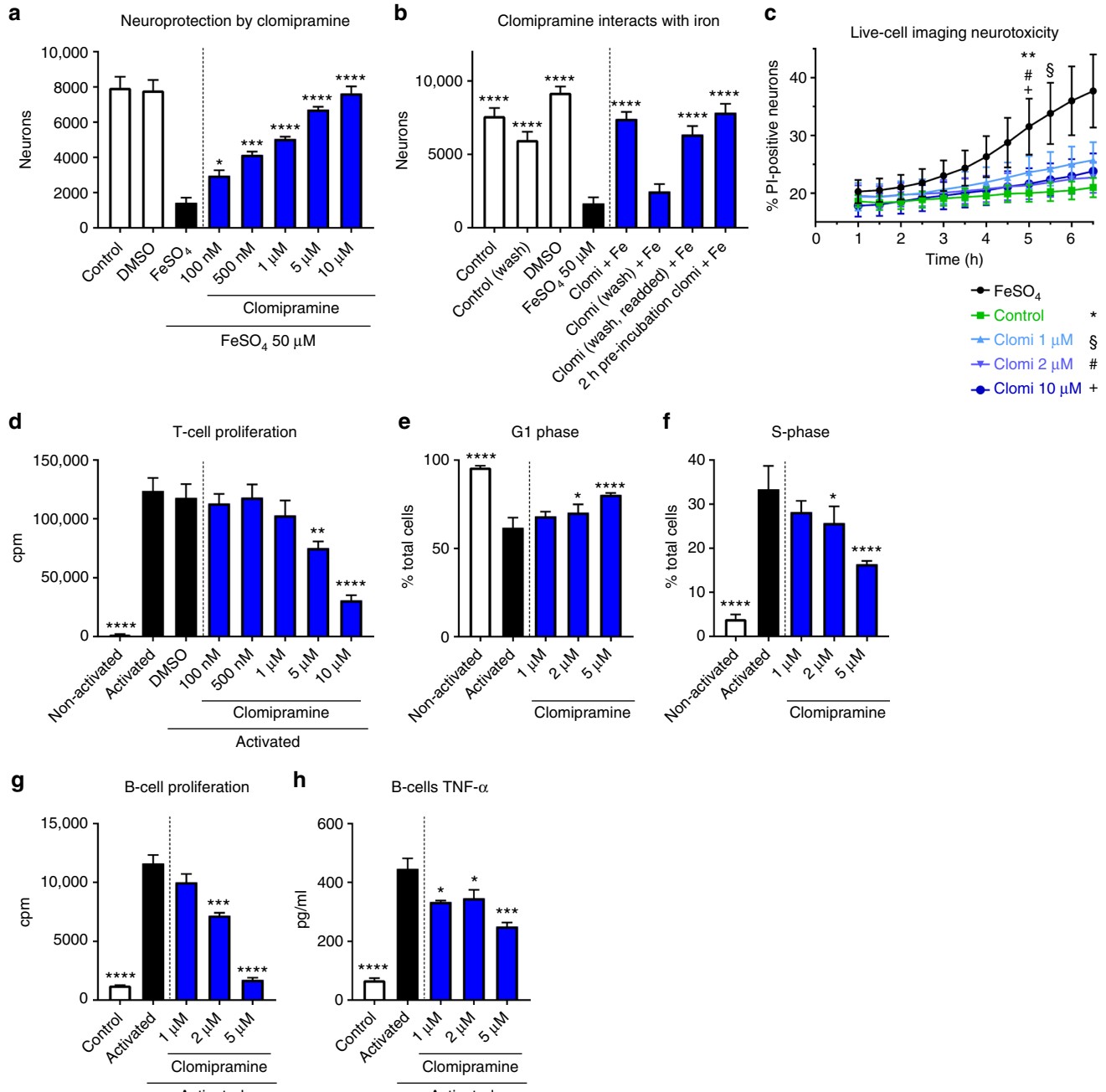

**Fig. 6** Clomipramine reduces iron neurotoxicity and proliferation of T-lymphocytes and B-lymphocytes. Clomipramine attenuated iron-mediated neurotoxicity in a concentration-dependent manner from 100 nM ($p < 0.05$) (**a**). Washing away clomipramine led to cell death by iron, but this effect could be prevented after pre-incubation of clomipramine with iron, suggesting a physical reaction between clomipramine and iron (**b**). Live-cell imaging studies show that the increasing accumulation of PI-positive neurons exposed to iron over time was prevented by clomipramine (**c**). Clomipramine furthermore reduced the proliferation of T-lymphocytes (**d**), reflected by a reduction of cells in S-phase and an increase in the G1-phase of the cell cycle (**e**, **f**). Proliferation of activated B-cells was reduced by clomipramine from 2 µM (**g**), correspondent with reduced TNF-α release (**h**). Data are shown as quadruplicate replicate wells of an individual experiment that was conducted twice (**a**, **d**, **e**, **f**), once (**b**) or three times (**g**, **h**); **c** represents triplicate wells of one experiment. Results are mean ± SEM. One-way ANOVA with Dunnett's multiple comparisons test as post hoc analysis compared to the FeSO4 or activated condition (**a**, **b**, **d–h**) and two-way ANOVA with Dunnett's multiple comparisons test (**c**): *$p < 0.05$; **$p < 0.01$; ***$p < 0.001$; ****$p < 0.0001$

Another model of chronic EAE, thought to model secondary progressive multiple sclerosis[22,23], is immunization with spinal cord homogenate (SCH) in the Biozzi ABH mouse. Clomipramine treatment was started at the onset of clinical signs where it reduced clinical severity throughout the period of treatment ($p = 0.0062$) (Fig. 10c).

In summary, clomipramine reduced clinical severity in acute and chronic EAE in two different mouse models. Figure 10d schematizes that the initiation of clomipramine treatment from onset of clinical signs of EAE attenuates the clinical disability observed during relapses or in chronic disease.

## Discussion

Unlike relapsing-remitting multiple sclerosis, trials in progressive multiple sclerosis have largely failed so far. One important

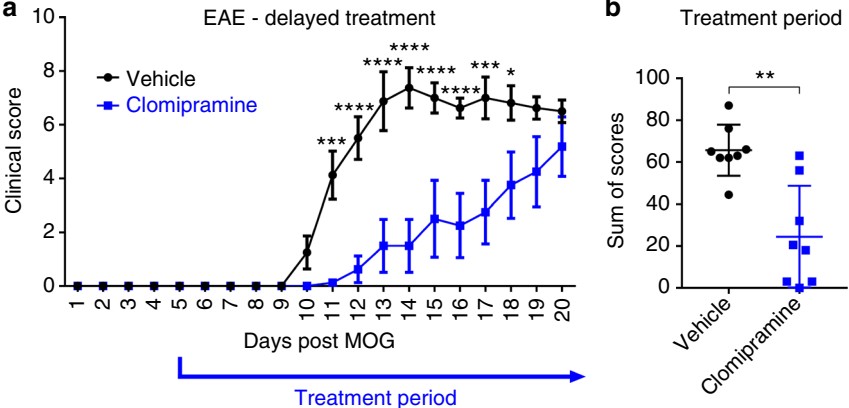

**Fig. 7** Clomipramine initiated from day 5 delays the onset of EAE clinical disease. Female C57BL/6 mice (age 8–10 weeks) were treated with clomipramine IP (25 mg/kg) or PBS (vehicle) from day 5 after induction of MOG-EAE (**a**). The disease onset was delayed and from day 11 the clinical course differed significantly ($p < 0.001$). Eventually, clomipramine-treated mice also developed the same disease burden as vehicle-treated mice. The overall disease burden is shown in (**b**). $n = 8$ vehicle and $n = 8$ clomipramine EAE mice. Data are depicted as mean ± SEM. Two-way ANOVA with Sidak's multiple-comparisons test as post hoc analysis (**a**) and two-tailed unpaired non-parametric Mann–Whitney test (**b**). Significance is shown as *$p < 0.05$; **$p < 0.01$; ***$p < 0.001$; ****$p < 0.0001$

explanation is the lack of directed actions of medications against features that drive the pathophysiology of progressive multiple sclerosis, and the lack of consideration of penetration of agents into the CNS. The latter is important as the blood–brain barrier appears relatively intact in progressive compared to the relapsing-remitting form[5], and pathogenic processes ongoing within the CNS may not be amendable to periphery-acting medications. To circumvent these challenges, we have employed bioassay screens that model aspects of progressive multiple sclerosis. Moreover, we have opted to test generic medications that have data of good access into the CNS.

One pathogenic hallmark important for the progression of multiple sclerosis is iron-mediated neurotoxicity. Iron accumulates in the CNS age dependently and iron deposition concomitant with T-cell infiltration and the expression of inducible nitric oxide synthase in microglia in the deep gray matter correlates with progression and is associated with neurodegeneration[24]. The deposition of iron amplifies inflammation and exacerbates mitochondrial dysfunction through oxidative stress, eventually leading to neurodegeneration[25]. Targeting iron is thus considered a promising therapeutic approach in progressive multiple sclerosis. Based on a screen of potential microglial inhibitors performed by our group[14], we investigated the potential of promising generic compounds to prevent iron-mediated neurotoxicity. Out of 249 compounds screened, 35 medications which prevented against iron-mediated neurotoxicity were in the drug classes of antidepressants ($n = 5$), antibiotics ($n = 4$), antipsychotics ($n = 3$), antimalarials ($n = 2$), and others. Some of the drugs had consistent outstanding neuroprotective effects, and these included antipsychotics and tricyclic antidepressants. The high number of antipsychotics and antidepressants as positive hits in the screening was striking. In addition to the rescue effect against iron-mediated neurotoxicity, several drugs showed promising results in other modes of toxicity; these were desipramine, clozapine, indapamide, and labetalol which were active against damage to the mitochondrial respiratory chain. Data were corroborated by the investigation of antioxidative potential and the influence on splenocyte proliferation. Clomipramine showed outstanding effects in several in vitro settings such as against iron-mediated neurotoxicity, hydroxyl scavenging capacity, and inhibition of T-cell and B-cell proliferation; in mice, clomipramine treatment suppressed occurrence of disease in EAE completely, concomitant with reduced transcripts of chemotactic and

inflammatory cytokines in the spinal cord, reduced inflammation, microglial activation, and preservation of axons. Moreover, clomipramine ameliorated clinical signs in chronic EAE in two different EAE models, C57BL/6 and Biozzi ABH mice.

The work presented here constitutes a systematic approach to identify generic compounds that could be useful for the treatment of progressive multiple sclerosis. First, we focused on ameliorating major hallmarks of progressive multiple sclerosis such as iron-mediated neurotoxicity, oxidative stress, and immune cell proliferation. Second, we chose generic drugs which are available as oral formulations. The drugs have a well-known safety profile, as there exists long-lasting experience in research and clinical use. Third, safety studies in patients with multiple sclerosis for these generic compounds will have to be conducted but they do not have to be exhaustive; thus, their translation into phase II and III clinical trials can be expedited.

The screening approach herein has limitations, which should be addressed. First, the screening was limited to a circumscribed number of generic medications contained within the NINDS library; it was fortunate that we found several hits to pursue. A second limitation is that due to the high number of compounds tested, only one concentration (10 μM) was investigated for the majority of compounds. This may lead to misses, as higher or lower concentrations of a particular drug may have to be administered in humans to reach effectiveness. A third limitation is the lack of a well-accepted animal model of progressive multiple sclerosis to test the promising drugs identified from the tissue culture studies. We utilized the acute EAE model as proof of concept for a drug's effectiveness in vivo followed by investigation in chronic EAE in C57BL/6 mice and Biozzi ABH EAE. While treatment from remission after the first relapse might not have been successful due to marked damage induced during the early phase of EAE, positive effects with early treatment during chronic disease in two models support potential effectiveness in human. Another limitation was the statistical analysis of the chronic experiments. Due to spontaneous remission of vehicle-treated animals, statistical significance would have been masked had the whole experiment been analyzed as a group, since vehicle-treated animals along with clomipramine-treated mice remitted to a very low disease score between days 25 and 42 (Fig. 10b). Hence, for statistical analyses, we focused on the differences of the acute and chronic relapse phases outside of the period of remission (Fig. 10b), as this would be more meaningful for a drug's utility in MS.

As no models of EAE are strongly suited for all aspects of progressive multiple sclerosis, we hope that the results of the current study will lead to a clinical trial in the ultimate test subjects: patients with progressive multiple sclerosis.

Another limitation of the screening approach was the sequence of the tests employed, beginning with amelioration of iron-

induced neurotoxicity to effects on T-cells. This sequence is not a reflection of which pathophysiology is most important for progressive multiple sclerosis, but simply because the protection of neurons against death in culture was an easily observed (through microscopy of living neurons) and therefore obvious outcome. Had the screen been sequenced in a different manner, other lead

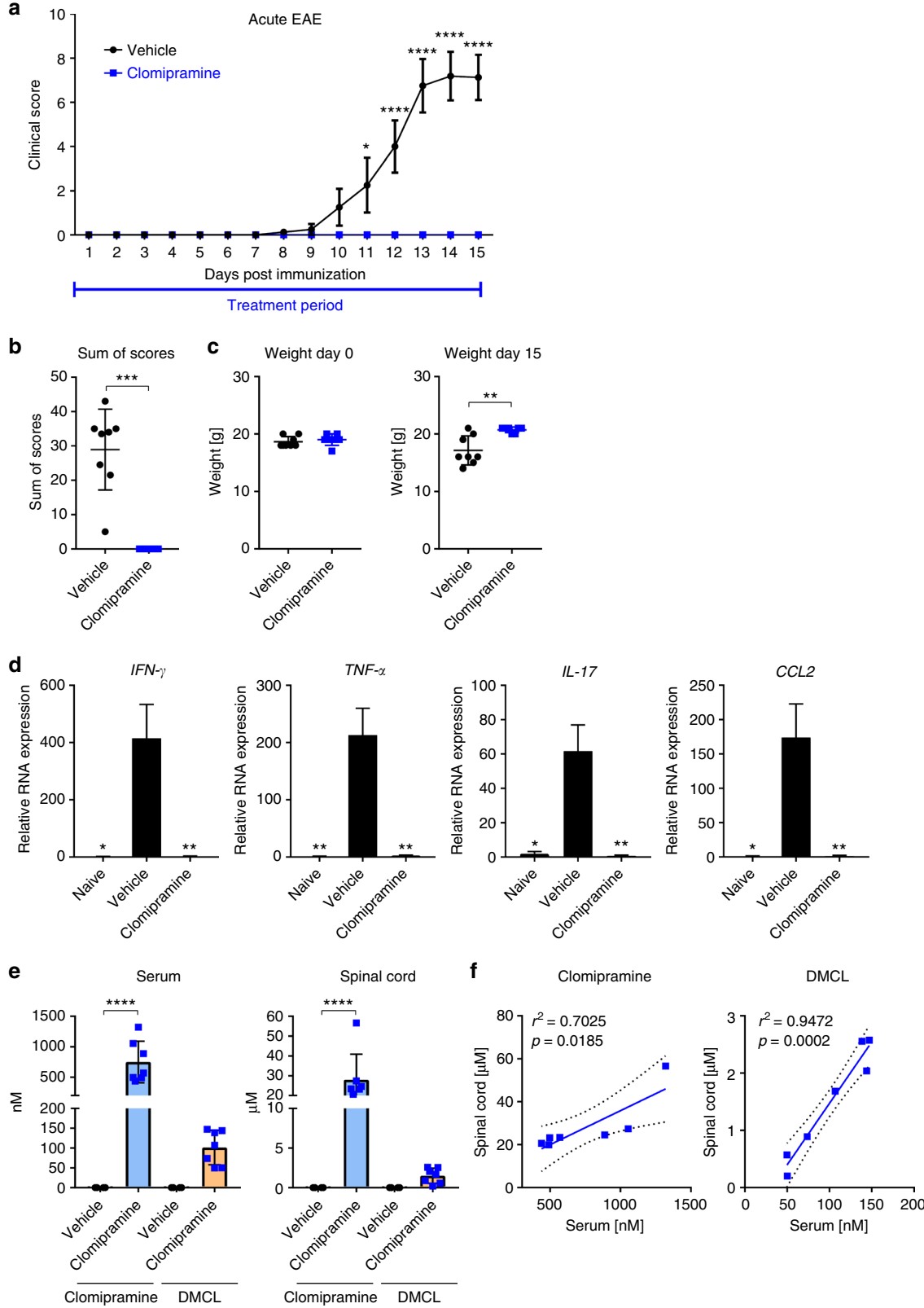

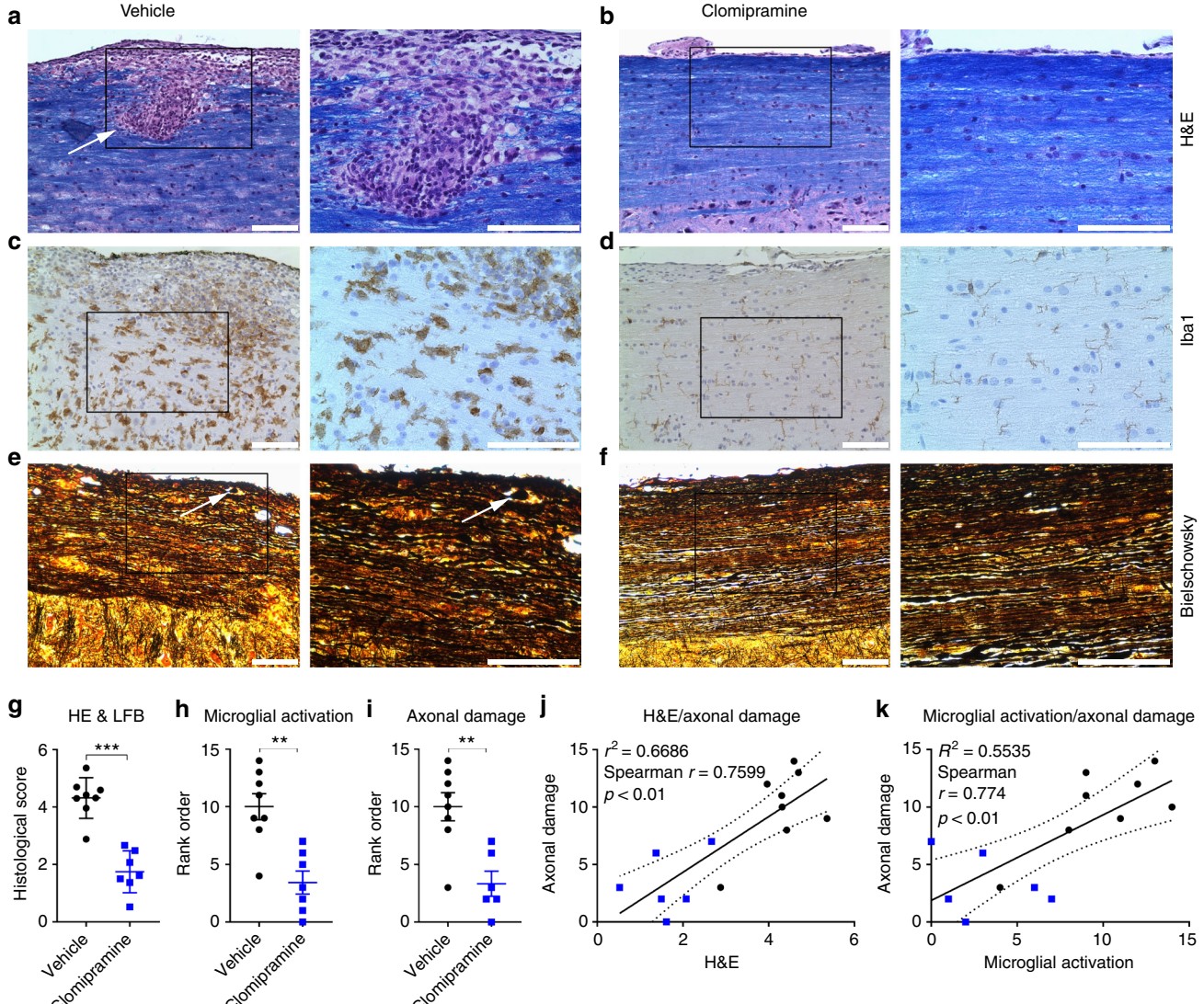

**Fig. 9** Reduced inflammation and axonal damage upon clomipramine treatment. Vehicle-treated animals had marked parenchymal inflammation, indicated by an arrow (**a**), whereas clomipramine-treated animals only had low meningeal inflammation (**b**). This was reflected in better histological scores (**g**) evaluated by a previously described method[54] (**a**, **b**: Hematoxylin/eosin and luxol fast blue, HE and LFB). Vehicle-treated animals had pronounced microglial activation (Iba1 stain, **c**), which was accompanied by axonal damage with formation of axonal bulbs (indicated by an arrow, Bielschowsky stain, **e**). Clomipramine treatment reduced microglial activation concomitant with preserved axonal integrity (**d**, **f**). This was reflected in a blinded rank order analysis (**h**, **i**). Infiltration and microglial activation positively correlated with axonal damage (**j**, **k**). **c**/**e** and **d**/**f** are adjacent sections. Images are shown in 20-times and 40-times original magnification. The scale bars show 100 μm. Non-parametric two-tailed Mann–Whitney test (**g**–**i**) and non-parametric two-tailed Spearman correlation with 95% confidence interval (**j**, **k**). Significance is shown as **$p < 0.01$; ***$p < 0.001$

**Fig. 8** Early clomipramine treatment suppressed EAE disease activity. Female C57BL/6 mice (age 8–10 weeks) were treated with clomipramine IP (25 mg/kg) or PBS (vehicle) from the day of induction of MOG-EAE (day 0). From day 11 the clinical course differed significantly ($p < 0.05$); while vehicle-treated mice accumulated progressive disability, clomipramine-treated mice remained unaffected even up to the termination of the experiment when vehicle-treated mice were at peak clinical severity (paralysis or paresis of tail and hind limb functions and paresis of forelimbs) (**a**). The overall burden of disease per mouse was plotted in **b**, while the relative weight of mice, reflecting general health, is shown in **c**. In the lumbar cord at animal sacrifice (day 15), there was a significant upregulation in vehicle-EAE mice of transcripts encoding *Ifng*, *Tnfa*, *Il-17*, and *Ccl2* compared to naïve mice, whereas clomipramine-treated mice did not show these elevations (**d**). Levels of clomipramine and the active metabolite desmethylclomipramine in serum and spinal cord at sacrifice (**e**) are consistent to concentrations reached in humans. There was a strong correlation of serum levels of clomipramine and desmethylclomipramine with spinal cord levels (**f**). Data in **d** are RT-PCR results, with values normalized to *Gapdh* as housekeeping gene and expressed in relation to levels in naïve mice. $n = 8$ vehicle and $n = 7$ clomipramine EAE mice. Data are depicted as mean ± SEM. Two-way ANOVA with Sidak's multiple-comparisons test as post hoc analysis (**a**), two-tailed unpaired non-parametric Mann–Whitney test (**b**), two-tailed unpaired *t*-test (**c**, **e**, **f**), and one-way ANOVA with Tukey's multiple comparisons test as post hoc analysis (**d**). Correlations were calculated using a linear regression model, dotted lines show the 95% confidence interval (**f**). Significance is shown as *$p < 0.05$; **$p < 0.01$; ***$p < 0.001$; ****$p < 0.0001$

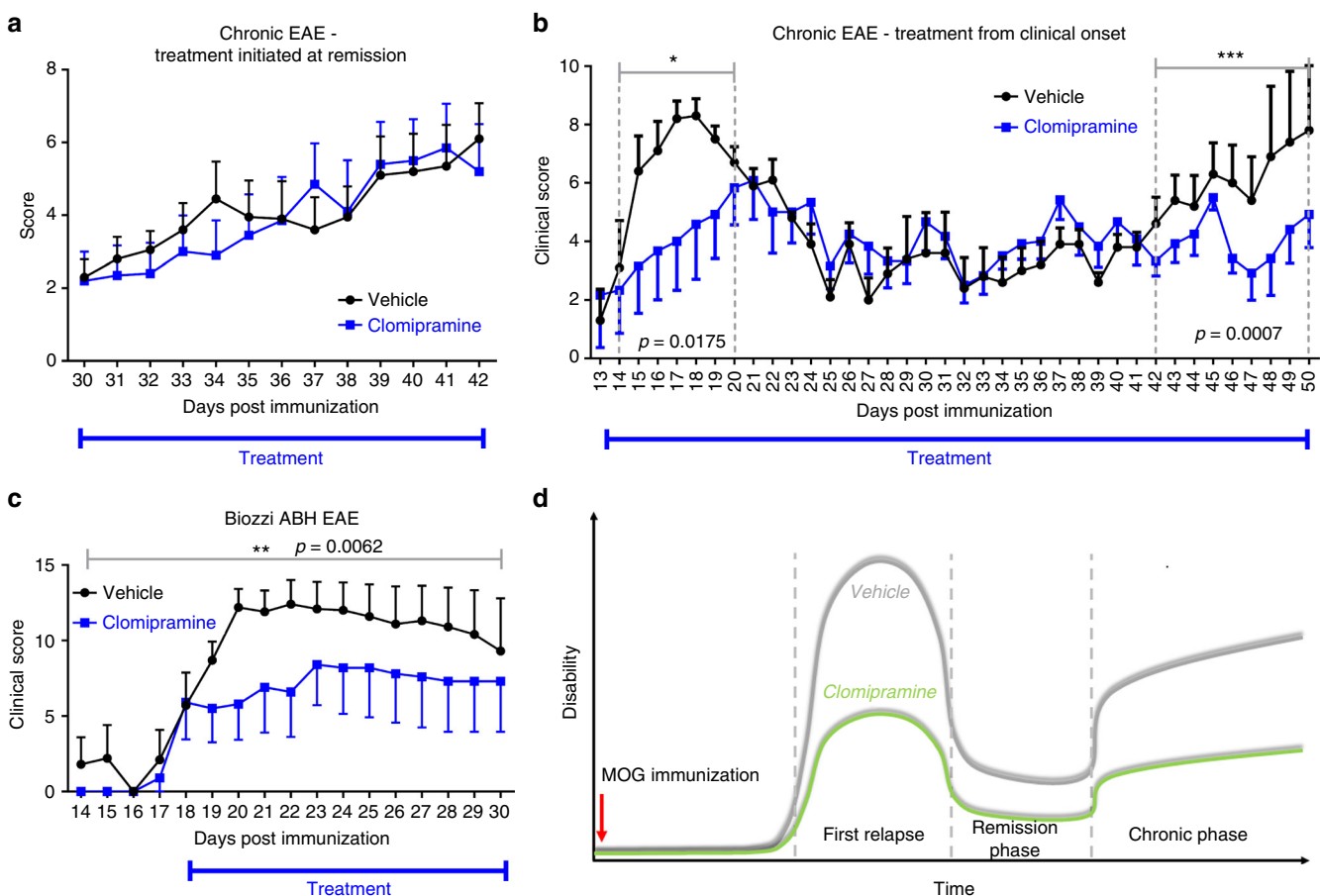

**Fig. 10** Clomipramine improves the chronic phase of EAE. **a** Female C57BL/6 (age 8–10 weeks) MOG-immunized mice were treated with clomipramine IP (25 mg/kg) or PBS (vehicle) from remission after the first relapse, and this did not affect disease score between the groups ($n = 10$ vehicle, $n = 10$ clomipramine). **b** In a second experiment, MOG-immunized C57BL/6 mice were treated from onset of clinical signs. Here, clomipramine reduced the clinical severity of the first relapse (day 14–20, $p = 0.0175$, two-tailed Mann–Whitney $t$-test) and of the second relapse at the late chronic phase (day 42–50, $p = 0.0007$, two-tailed Mann–Whitney $t$-test) ($n = 5$ vehicle, $n = 6$ clomipramine). Note that an initial two-way ANOVA with Sidak's multiple-comparisons test of the experiment from day 13–50 was not statistically significant, since vehicle-treated mice spontaneously remitted to a very low disease score between days 25 and 42, so that differences with the treatment group could not be detected. Hence, we analyzed differences of the acute and chronic relapse phases outside of the period of remission, using Mann–Whitney $t$-test. **c** Using Biozzi ABH mice, treatment from onset of clinical disability showed a positive effect on the chronic phase ($p = 0.0062$, two-tailed Mann–Whitney test) ($n = 5$ vehicle, $n = 5$ clomipramine). When a two-way ANOVA with Sidak's multiple-comparisons test was used, the results were not significant since the individual variability of mice in either group in any given day was very high for this model in our hands. **d** A summary of the effect of clomipramine when treatment is initiated at the onset of clinical signs

candidates might have emerged. Nonetheless, through the sequence that we employed, a testable medication in patients with multiple sclerosis, clomipramine, has emerged. Moreover, although a larger library could have been used, the NINDS collection was the largest grouping of generic medications available at the time of our study. We reiterate our focus on generics as their potential to advance faster into clinical studies is much higher, given that there would be knowledge of their spectrum of side effects in human use.

Some of the compounds that prevented iron-mediated neurotoxicity in our screen have been described previously to have neuroprotective properties and will be highlighted here, as they may be of interest not only to progressive multiple sclerosis but also other CNS disorders with neurodegenerative features. Strong neuroprotective effects were induced by tricyclic antidepressants. The antidepressant desipramine has been used in a Huntington's disease model where it inhibited glutamate-induced mitochondrial permeability at the concentration of 2 μM and led to reduced apoptosis of primary murine neurons[26,27]. Furthermore, desipramine induces the antioxidative enzyme heme-oxygenase 1 in Mes23.5 dopaminergic cells and increases Nrf2 accumulation in

the nucleus, thus preventing neuronal cell death mediated by rotenone and 6-hydroxydopamine[28]. We could not confirm the inhibition of rotenone-mediated cell death in our study of primary human neurons and we did not choose to pursue desipramine further as it promoted the formation of hydroxyl radicals in the HORAC assay. In murine EAE, desipramine reduces CCL5 in cortical homogenates and positively affects anxiety-related behavior but not clinical signs[29]. Of interest, efficacy of desipramine in depression in multiple sclerosis has been shown in a Cochrane analysis and was associated with fewer adverse effects than paroxetine[30].

Besides desipramine, other tricyclic antidepressants had strong effects against splenocyte proliferation. Imipramine, which showed good neuroprotective properties, enhances PEP-1-catalase in astrocytes, leading to neuroprotection in the hippocampal CA1 region in an ischemia model[31]. Additionally, it prevents apoptosis of neural stem cells by lipopolysaccharide, mediated by the brain-derived neurotrophic factor and mitogen-activated protein kinase pathway[32]. Another novel compound recently developed, quinpramine, which is a fusion of imipramine and the antimalarial quinacrine, decreased the number of

inflammatory CNS lesions, antigen-specific T-cell proliferation and pro-inflammatory cytokines in EAE[33].

Due to structural similarities between clomipramine, imipramine, and trimipramine it may be speculated that these compounds may be relevant for trials in progressive multiple sclerosis. Furthermore, we showed previously that doxepin reduces microglial activation to 46% without inducing toxicity; clomipramine, however, did not have microglia inhibitory activity[14]. In the synopsis of effects contributing to progressive multiple sclerosis, tricyclic antidepressants are interesting for further development and might even be suitable as combination therapy with other compounds targeting features of progressive multiple sclerosis.

Some antipsychotics also displayed strong protection against iron and oxidative stress. Clozapine has been described to reduce microglial activation through inhibition of phagocytic oxidase-generated reactive oxygen species production, mediating neuroprotection[34]. The strong antioxidative properties of clozapine in the HORAC assay support these results. Due to the side effect profile with enhanced risk of agranulocytosis, we refrained from usage in EAE; nevertheless, in multiple sclerosis patients with psychiatric comorbidities and eligible for antipsychotic treatment, it may be reasonable to use clozapine.

With regard to liothyronine, atenolol or carvedilol that prevented iron-mediated neurotoxicity beyond levels of controls, these do not penetrate the CNS (probability of 68% for all three, drugbank.ca) as well as clomipramine (97.9% chance for entering the CNS according to drugbank.ca). Thus, we did not explore their utility in EAE.

Mitoxantrone is used in some countries as a treatment for progressive multiple sclerosis, but has so far not yet been described as being neuroprotective. Although the blood–brain barrier permeability probability is poor (0.7979), it may be postulated that the effect in progressive multiple sclerosis, in addition to its toxic effects on T-lymphocytes, is induced by its capacity to limit iron-mediated neurotoxicity. Indapamide exhibited strong neuroprotective effects against iron toxicity in culture, which has not yet been described previously. More interestingly, indapamide also overcomes mitochondrial damage. As indapamide has no effect on T-lymphocyte proliferation, the drug may not overcome acute-EAE, but may be interesting in longer-term multiple sclerosis models such as the Biozzi ABH mouse model, which shows immune cell-independent neurodegeneration[35] and a chronic disease course[22].

We opted to test clomipramine in the acute-EAE model due to its strong effects on immune cells, its antioxidative properties and its prevention against iron-mediated neurotoxicity. Clomipramine is a tricyclic antidepressant which is used to treat depression, obsessive compulsive disorder and panic disorders, usually in a dosage of 100–150 mg/d, sometimes up to 300 mg/d. It inhibits serotonin and norepinephrine uptake. Clomipramine reduces the seizure threshold and overdose can lead to cardiac dysrhythmias, hypotension, and coma (drugbank.ca). Usually, clomipramine is well tolerated, but side effects include among others increase in weight, sexual dysfunctions, sedation, hypotension and anticholinergic effects such as dry mouth, sweating, obstipation, blurred vision, and micturition disorder (according to the manufacturer leaflet). Clomipramine crosses readily into the CNS with a probability to cross the blood–brain barrier of 0.979 according to predicted ADMET (absorption, distribution, metabolism, excretion, toxicity) features (drugbank.ca). Clomipramine reduces the production of nitric oxide and TNF-α in microglia and astrocytes[36]; the authors reported neuroprotective properties in a co-culture model of neuroblastoma cells and microglia. Clomipramine increases the uptake of cortisol in primary rat neurons[37] and promotes the release of glial cell line-derived neurotrophic factor in glioblastoma cells, suggesting a protective effect on neurons[38]. The drug has been also studied in experimental autoimmune neuritis, where it

decreases the number of IFN-γ secreting Th1 cells and ameliorated the clinical course[39].

Clomipramine has been used previously in mice in different dosages to study conditions such as antinociception (0.5 mg/kg)[40], Chagas disease (7.5 mg/kg)[41], and neurotransmitter and histone deacetylase expression (50 mg/kg)[42]. In humans taking clomipramine as an antidepressant, mean serum levels after a mean daily intake of $127 \pm 91$ mg/d have been reported to be 122 ng/ml (387 nM, considering a molecular weight of 314.9)[17]. Of note, clomipramine levels after oral intake in humans have a wide range, leading to plasma concentrations of more than 600 nM in some individuals[18], which is in the range of neuroprotection against iron in our in vitro experiments. The injection of 20 mg/kg IP in CD1 mice leads to peak plasma concentrations of 438 ng/ml (1.4 μM) with a half-life of 165 min[21] and in our experiments animals (sacrificed 1 h after the last injection) had mean serum clomipramine concentrations of 236.5 ng/ml (751 nM). These plasma levels are close to the ones measured in humans (average of 387 nM, and up to 600 nM[18]), especially keeping in mind that plasma levels drop faster in mice due to the relatively bigger liver: body mass and that the half-life of clomipramine in humans is between 17.7 and 84 h[43] compared to about 2.5 h in mice. We found that clomipramine levels in the spinal cord of the EAE-afflicted mice averaged an incredibly high 28 μM; levels achieved in the brains of humans are not known. Thus, the dosage of 25 mg/kg clomipramine tested in our EAE study reflects standard doses used in humans in that both attain similar plasma levels.

In summary, we discovered several generic compounds in this systematic screening approach that exhibit neuroprotective properties against iron-mediated neurotoxicity. Additionally, some of those compounds prevent mitochondrial damage to neurons, inhibit immune cell proliferation, and show anti-oxidative capacities. They may thus be interesting for further development in progressive multiple sclerosis, and other neurodegenerative diseases such as Alzheimer's dementia, Huntington's and Parkinson's disease. Tricyclic antidepressants, antipsychotics, and indapamide may be useful for further development in progressive multiple sclerosis due to their manifold properties. Clomipramine showed particular promise due to its capacity to reduce iron-mediated neurotoxicity and T-cell and B-cell proliferation, its antioxidative effect, and its complete suppression of disease in acute-EAE and positive effects in chronic EAE. As these features are relevant in progressive multiple sclerosis, we propose clomipramine for a trial in the unmet need of progressive multiple sclerosis.

## Methods

**Cell culture and treatment of human neurons**. Human neurons were isolated from brain tissues of therapeutically aborted 15–20-week-old fetuses, in accordance with ethics approval of the University of Calgary ethics committee, after written informed consent of the pregnant donors. Neurons were isolated as follows in brief[44]: brain specimens were washed in phosphate buffered saline (PBS) to remove blood, followed by removal of meninges. Tissue was mechanically dissected, followed by digestion in DNase (6–8 ml of 1 mg/ml; Roche), 4 ml 2.5% trypsin and 40 ml PBS (37 °C, 25 min). Thereafter, the digestion was stopped by addition of 4 ml fetal calf serum. The solution was filtered through a 132 μm filter and centrifuged (three times, 1200 rpm, 10 min). Cells were cultured in feeding medium of minimal essential medium supplemented with 10% fetal bovine serum (FBS), 1 μM sodium pyruvate, 10 μM glutamine, 1× non-essential amino acids, 0.1% dextrose, and 1% penicillin/streptomycin (all culture supplements from Invitrogen, Burlington, Canada). The initial isolates of mixed CNS cell types were plated in poly-L-ornithine-coated (10 μg/ml) T75 flasks and cultured for at least two cycles[44] in medium containing 25 μM cytosine arabinoside (Sigma-Aldrich, Oakville, Canada) to inhibit astrocyte proliferation and to deplete this major contaminating cell type. For experiments, the neuron-enriched cultures were re-trypsinized and cells were plated in poly-L-ornithine pre-coated 96-well plates at a density of 100,000 cells/well in 100 μl of the complete medium supplemented with cytosine arabinoside. Medium was changed to AIM V® Serum Free Medium (Invitrogen) after 24 h. After a period of 1 h, respective drugs were added in a concentration of 10 μM, followed by application of $FeSO_4$ or rotenone after 1 h. All conditions were

performed in quadruplicates. A day later cells were fixed using 4% paraformaldehyde and stored in PBS at 4 °C.

We note that in tissue culture, the toxicity of iron to neurons begins immediately. Thus, it has been our experience that pretreatment with test protective agents is necessary. With the continuous insult that occurs in multiple sclerosis, a pretreatment paradigm with test compounds against iron neurotoxicity in our experiments can be justified as that simulates the protection against the next injury in the disease.

Drugs tested were contained within the 1040-compound NINDS Custom Collection II, which was purchased from Microsource Discovery (Gaylordsville, CT, USA) and used as previously described[14]. Briefly, there were 80 compounds located in specific wells on each plate (e.g., B07). 3B07 would thus refer to position B07 of plate 3. Each compound was supplied at a concentration of 10 mM dissolved in dimethyl sulfoxide (DMSO).

The iron stock solution was prepared using 27.8 mg iron(II) sulfate heptahydrate (FeSO$_4$) (Sigma-Aldrich, Oakville, Canada), 10 µl of 17.8 M sulfuric acid, and 10 ml deionized distilled water. After filtering with a 0.2 µm filter, FeSO$_4$ was added to cells in a final concentration of 25–50 µM in a volume of 50 µl medium to the cells. Rotenone was dissolved in DMSO and used in a final concentration of 10 µM.

**HORAC assay**. Selected compounds that prevented iron-mediated neurotoxicity were analyzed for their antioxidative properties using the HORAC assay, in accordance with the procedure outlined in Číž et al[45]. In this assay, hydroxyl radicals generated by a Co(II)-mediated Fenton-like reaction oxidize fluorescein causing loss of fluorescence[46]. The presence of an antioxidant reduces the loss of fluorescence and this can be monitored every 5 min over a period of 60 min with a Spectra Max Gemini XS plate reader (Molecular Devices, Sunnyvale, CA, USA) and the software SoftMax Pro version 5. For monitoring fluorescence, we used an excitation wavelength of $\lambda = 485$ nm and an emission wavelength of $\lambda = 520$ nm.

**Proliferation of T-lymphocytes**. A previously published protocol was used for isolating and activating T-cells[47]. Spleens from female C57BL/6 mice were harvested and after mechanical dissociation the cell suspension was passed through a 70 µm cell strainer and separated by Ficoll gradient (1800 RPM, 30 min). Splenocytes were plated ($2.5 \times 10^5$ cells in 100 µl/well) in anti-CD3 antibody-coated 96-well plates (1000 ng ml$^{-1}$ plate-bound anti-CD3 and 1000 ng ml$^{-1}$ anti-CD28 suspended in media) to activate T-cells. Directly before plating, wells were treated with respective drugs in a final concentration of 10 µM. Cells were cultured in RPMI 1640 medium, supplemented with 10% FBS, 1 µM sodium pyruvate, 2 mM L-alanyl-L-glutamine, 1% penicillin/streptomycin, 1% HEPES, and 0.05 mM 2-mercaptoethanol (all supplements were from Invitrogen). After 48 h, $^3$H-thymidine was added in a concentration of 1 µCi per well, and cells were harvested after 24 h on filter mats. Mats were then evaluated for radioactivity (counts per minute, cpm) using a liquid scintillation counter.

**Activity on B-lymphocytes**. Venous blood from healthy volunteers was obtained and peripheral blood mononuclear cells (PBMCs) were isolated by Ficoll gradient centrifugation (1800 RPM, 30 min). From PBMCs, B-cells were isolated by positive selection with CD19-directed microbeads (Stemcell Technologies). Purity was assessed by FACS after staining for CD19 (Stemcell Technologies). Cells were plated at a concentration of $2.5 \times 10^5$ cells/well in X-VIVO™ medium (Lonza) supplemented with 1% penicillin/streptomycin and 1% Glutamax and treated with drugs for 1 h. Cells were then activated with 10 µg/ml IgM BCR cross-linking antibody (XAb) (Jackson ImmunoResearch), 1 µg/ml anti-CD40L and IL-4 20 ng/ml for 24 h as described previously[48]. Conditioned media were harvested after 24 h for ELISA. Medium as well as respective drugs were re-added followed by application of $^3$H-thymidine in a concentration of 1 µCi per well to investigate proliferation. After 24 h, cells were harvested on filter mats and after drying cpm were measured using a liquid scintillation counter.

**Flow cytometry**. Two days after activation and drug treatment splenocytes were harvested, washed with PBS followed by resuspension in PBS with 2% FBS. Cell cycle analysis was performed taking advantage of PI staining using an established protocol[49]. Cells were washed in cold PBS and resuspended in PI/Triton X-100 staining solution (10 ml 0.1% (v/v) Triton X-100 in PBS with 2 mg DNAse-free RNAse A and 0.4 ml of 500 µg/ml PI), followed by incubation at 4 °C for 30 min. Stained cells were analyzed on a FACSCalibur™ with the software CellQuest™ (BD Biosciences). Cell cycle analysis was conducted using the software ModFit LT, version 3.3 (Verity Software House Inc.). Cells were identified by gating into the lymphocyte population, followed by single cell gating to exclude doublets and aggregates. This was followed by identification of the G0/G1 population and processing with the software ModFit LT, version 3.3 (Verity Software House Inc.) to calculate the percentage of cells in different cell cycles.

**Immunocytochemistry and microscopy**. Staining was performed at room temperature. A blocking buffer was first introduced for 1 h followed by incubation with primary antibody overnight in 4 °C. Neurons were stained using mouse anti-MAP-2 antibody, clone HM-2 (dilution 1:1000; Sigma-Aldrich, catalog number M4403,

species mouse). Primary antibody was visualized with Alexa Fluor 488 or 546-conjugated goat anti-mouse secondary antibody (dilution 1:250, Invitrogen, Burlington, Canada). Cell nuclei were stained with Hoechst S769121 (nuclear yellow). Cells were stored in 4 °C in the dark before imaging.

Images were taken using the automated ImageXpress® imaging system (Molecular Devices, Sunnyvale, CA) through a 10× objective microscope lens, displaying 4 or 9 sites per well. Images were analyzed with the software MetaXpress® (Molecular Devices, Sunnyvale, CA) using the algorithm "multiwavelength cell scoring"[50]. Cells were defined according to fluorescence intensity and size at different wavelengths. Data from all sites per well were averaged to one data point.

**Live-cell imaging**. Neurons were prepared as described above. Directly after the addition of FeSO$_4$ to healthy neurons, the live cell-permeant Hoechst 33342 (1:2 diluted in AIM-V medium, nuclear blue; Thermo Fisher Scientific, Grand Island, NY, USA) and the live cell-impermeable PI (1:20 diluted in AIM-V medium) were added in a volume of 20 µl (Sigma-Aldrich). In compromised cells, PI could now diffuse across the plasma membrane. Live-cell imaging was performed using the automated ImageXpress® imaging system under controlled environmental conditions (37 °C and 5% CO$_2$). Images were taken from nine sites per well at baseline and then every 30 min for 12 h. After export with MetaXpress®, videos were edited with ImageJ (NIH) in a uniform manner. Nuclei were pseudo colored in cyan, PI-positive cells in red.

**Experimental autoimmune encephalomyelitis**. EAE was induced in 8–10-week-old female C57BL/6 mice (Charles River, Montreal, Canada). Mice were injected with 50 µg of MOG$_{35-55}$ (synthesized by the Peptide Facility of the University of Calgary) in Complete Freund's Adjuvant (Thermo Fisher Scientific) supplemented with 10 mg/ml Mycobacterium tuberculosis subcutaneously on both hind flanks on day 0. In addition, pertussis toxin (300 ng/200 µl; List Biological Laboratories, Hornby, Canada) was injected intraperitoneal (IP) on days 0 and 2. Animals were treated with clomipramine (25 mg/kg; 100 µl of 5 mg/ml solution) by IP injection from day 0 or day 5 (Figs. 7, 8), from day 30 at remission (Fig. 10a), or from 13 at onset of clinical signs (Fig. 10b). The solution of clomipramine was prepared daily in PBS.

The Biozzi ABH mouse model[22] was used as a model of progression. EAE was induced in Biozzi ABH mice aged 8–10 weeks by the subcutaneous application of 150 µl emulsion in both sides of the hind flanks[22]. The emulsion was prepared as follows: Stock A consisted of 4 ml of incomplete Freund's adjuvant mixed with 16 mg M. tuberculosis and 2 mg M. butyricum. 1 ml of stock A was mixed with 11.5 ml incomplete Freund's adjuvant to become stock B. Stock B was mixed in equal volume with SCH in PBS before injection. SCH was used in a concentration of 6.6 mg/ml emulsion each for 2 injections (days 0 and 7).

The number of animals was chosen according to experience with previous experiments (Fig. 7: 8/8 (vehicle/clomipramine); Fig. 8: 8/7; Fig. 10a 10/10; b 5/6; c 5/ 5), and animals were randomized after induction of EAE. Animals were handled according to the Canadian Council for Animal Care and the guidelines of the animal facility of the University of Calgary. All animal experiments received ethics approval (AC12–0181) from the University of Calgary's Animal Ethics Committee. Mice were scored daily using a 15-point scoring system, the investigator was not blinded[51].

**Histological analyses**. 1 h after the last administration of clomipramine animals were anesthetized with ketamine/xylazine, blood was taken by an intracardiac puncture for serum, and animals were then subjected to PBS-perfusion. Spinal cords and cerebella were removed. The thoracic cords were fixed in 10% buffered formalin, followed by embedding in paraffin. Cervical and lumbar cords were snap frozen. Tissue was further processed as previously described[52]. Briefly, the thoracic spinal cord was cut longitudinally from the ventral to the dorsal side with sections of 6 µm thickness. Sections were stained with hematoxylin/eosin, Iba1 (dilution 1:250; Wako, catalog number 019-18741, species rabbit) to visualize microglia and Bielschowsky's silver stain to visualize axons. Sections for Iba1 and Bielschowsky's silver stain were blinded, before images depicting area of maximal microglial activation or axonal damage were chosen for blinded rank order analysis by a second investigator.

**PCR**. Lumbar spinal cords were harvested, snap frozen in liquid nitrogen and stored in −80 °C. Samples were homogenized in 1 ml Trizol followed by the addition of 200 µl chloroform. The suspension was shaken, centrifuged (11,500 RPM for 15 min at 4 °C) and the RNA-containing upper phase was transferred into a new tube and precipitated with equal amounts of 70% ethanol. RNA was extracted using the RNeasy Mini Kit according to the manufacturer's instruction (Qiagen). RNA concentrations were measured using a Nanodrop (Thermo Fisher Scientific). cDNA preparation was performed using the RT$^2$ First Strand kit (Qiagen) with 1 µg of RNA according to the manufacturer's instructions. Real-time PCR was performed using the QuantStudio 6 Flex (Applied Biosystems by Life Technologies) with FAST SYBR Green and primers for *Gapdh* (Qiagen) as housekeeping gene, *Ifn-γ* (Qiagen, QT01038821), *Tnfa* (Qiagen, QT00104006), *Il-17* (SABiosciences, PPM03023A-200), and *Ccl2* (Qiagen, QT00167832). Relative

expression was calculated using the ΔΔCT method with *Gapdh* as housekeeping gene. Data were normalized to gene expression in naïve mice.

**Liquid chromatography-mass spectrometry (LC-MS)**. The assay is a modification of the LC-MS assay of Shinokuzack et al[53]. For preparation of samples, 100 µl of ice cold methanol were added to 100 µl of serum in each sample after addition of the internal standard maprotiline. The tubes were vortexed and left on ice for 10 min followed by centrifugation at 10,000 x *g* for 4 min. An equal amount of distilled water was added to each supernatant. Spinal cord samples were each homogenized in 10 volumes of ice-cold 80% methanol. 20 µl of o-phosphoric acid were added to all samples after addition of internal standard (maprotiline). The tubes were vortexed and left on ice for 10 min, followed by centrifugation at 10,000 x *g* for 4 min and an equal volume of distilled water was added to each supernatant.

An HLB Prime µelution plate was employed for sample cleanup for both serum and spinal cord samples. After running the supernatants described above through the wells, all wells were washed with 5% methanol in water and allowed to dry completely before elution with 100 µl 0.05% formic acid in methanol: acetonitrile (1:1). The eluents were transferred to low volume µl glass inserts (Waters, Milford, MA, USA) and 10 µl from each eluent were injected into the LC-MS system.

Analyses were conducted with a ZQ Mass detector - 2695 Separations module from Waters. The control of instrument, and the acquisition and processing of results were enabled by Mass Lynx 4.0 software. HPLC runs were processed by a 3 µm, 3.0 × 100 mm Atlantis dC18 column with similar material for the guard column. The mobile phases A and B were 0.05% formic acid constituted in water and acetonitrile respectively. The procedure started with a flow rate of 0.3 mL/min comprising of 80% of A and 20% of B. This was elevated in 15 min to 80% B and then restituted to starting conditions. The sample cooler and column heater were maintained at 4 and 30 °C, respectively. The optimized positive electrospray parameters had specifications of 1.2V Rf lens voltage; 3.77 kV capillary voltage; 110 °C source; 300 °C desolvation temperature; 80 L/h cone gas flow (nitrogen) and 300 L/h of desolvation nitrogen gas flow. Cone voltage was varied for each compound: clomipramine 25 V; N-desmethylclomipramine 22 V; and maprotiline 25 V. The *m/z* ratios for clomipramine, N-desmethylclomipramine, and maprotiline (internal standard) were 315, 301, and 278, respectively.

Calibration curves consisting of varying amounts of authentic clomipramine and N-desmethylclomipramine and the same fixed amount of maprotiline as added to the samples being analyzed were run in parallel through the procedure described above and the ratios of clomipramine and N-desmethylclomipramine to maprotiline were used to determine the amount of drug and metabolite in the serum and spinal cord samples.

**Statistical analysis**. Statistical analysis was performed using the Graphpad Prism software version 7 (La Jolla, CA, USA). For cell culture experiments, one-way ANOVA with different post hoc analyses was applied, as stated in the respective figure legends. EAE scores were analyzed using two-way ANOVA with Sidak's multiple comparison as post hoc analysis or non-parametric two-tailed Mann–Whitney test. Sum of scores and rank order analysis were analyzed with the non-parametric two-tailed Mann–Whitney test. Animal weight and drug levels were analyzed with an unpaired two-tailed *t*-test. Correlations of drug levels were calculated using a linear regression model. Correlations of rank order analyses were performed using non-parametric two-tailed Spearman correlation. Statistical significance was considered as $p < 0.05$ (*), $p < 0.01$ (**), $p < 0.001$ (***), and $p < 0.0001$ (****). All experiments were performed in quadruplicates, if not otherwise specified.

**Data availability**. All relevant data are available from the authors upon reasonable request.

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

## Acknowledgements

The authors are grateful to Drs. Glen Baker and Christopher Power at the University of Alberta for oversight of the assays for levels of clomipramine and N-desmethylclomipramine, and for helpful discussions. We thank the laboratory of Dr. Amit Bar-Or, McGill University, for guidance of B-cell cultures. The authors acknowledge use of the microscopy "RUN CORE" facility of the Hotchkiss Brain Institute, University of Calgary. This study was funded by an Alberta Innovates—Health Solutions CRIO Team program and the Hotchkiss Brain Institute Multiple Sclerosis Translational Clinical Trials Research Program. S.F. is supported by a research price of the Medical Faculty of the Ruhr-University Bochum and the Foundation for Therapeutic Research. V.W.Y. is supported by a Canada Research Chair (Tier 1) award.

## Author contributions

S.F. designed and conducted experiments, acquired and analyzed data, drafted figures and wrote the manuscript; M.M. acquired and analyzed data and critically revised the manuscript; D.K. conducted experiments and analyzed data; J.W., Y.F., G.R., and C.S. conducted experiments and analyzed data; L.M. designed experiments and critically revised the manuscript; M. K. designed experiments, analyzed data and critically revised the manuscript; V.W.Y. designed experiments, analyzed data, supervised the overall study and wrote the manuscript.

## Additional information

**Competing interests:** S.F., M.K., and V.W.Y. filed a provisional patent application at the US FDA. The remaining authors declare no competing financial interests.

