## [Peer Review File · Nature Communications]

Reviewers' comments:

Reviewer #1 (Remarks to the Author):

Reviewer comments regarding the manuscript 'Systematic screening of generic drugs for progressive multiple sclerosis: 1 Clomipramine as a promising therapeutic' submitted to Nature Communications.

In this experimental study, the authors established a high-throughput screening assay to identify novel potential pharmaceutical compounds for the treatment of progressive multiple sclerosis (MS). They based their screening on measures of neurotoxicity, mitochondrial damage, reactive oxygen species, and T cell proliferation. Subsequently, they used this assay to screen a large drug collection and identify tricyclic anti-depressants as promising new drug candidates. From this class of drugs the authors chose clomipramine to demonstrate the efficacy in an animal model of MS.

Overall this is an excellently written and very well performed experimental study that establishes a novel screening approach and identifies a novel class of compounds. On the one hand, the study thus creates the necessary tools to screen and validate novel compounds for the future treatment of a debilitating human disorder. On the other hand, the study sets the stage for testing an entire class of drugs in progressive MS. A speculative immediate implication of this study could be to preferentially use tricyclic rather than other types of antidepressants for the treatment of depression in progressive MS patients. Overall, this is a highly important, exciting and innovative study and its findings clearly deserve publication and further investigation.

Major points:

What should be better explained is, how the prioritization of screening approaches is to be understood, and how the innovative and original screening can be better linked to a good in vivo model for plausibility of the mechanisms linked.

The authors emphasize the applicability of their screening approach to chronic MS. But then they use MOG35-55 peptide induced EAE in C57BL/6 mice and use a follow-up of 19 and 15 days respectively to illustrate the applicability of Clomipramine in MS. The use of this specific animal model in this context can be discussed critically. If at all, this EAE variant is a model of one acute MS relapse especially if followed-up for such brief periods of time. Axonal loss in MOG peptide induced EAE correlates closely with peak disease severity and this does not prove a neuroprotective effect of the compound. Following the animals for only 15 days (Fig. 8) has limited value given that the peak severity of EAE seems to be delayed rather than suppressed in Figure 7. Various models have been developed that reproduce some aspects of chronic MS e.g. actively induced EAE in Biozzi ABH mice (as discussed in line 338), in NOD mice (used e.g. in Mayo et al. 2015 Nat Med), in DA rats, in SJL mice or spontaneous models such as the 2D2xTH MOG TCR and BCR double transgenic mouse line. Overall, the second part of this study shows a clear drop in quality and does not support using Clomipramine in chronic MS at all. So, the authors are advised to use and implement at least one additional model, that better mimics aspects of „progressive“ nature of the disease

Part of the authors' selection criteria for compounds was oral bio-availability (line 111). In a real world scenario this would eliminate for example Ocrelizumab from a list of potential compounds and there is no good reason to believe that an effective treatment for chronic MS would HAVE to be oral. This narrows the list of compounds unnecessarily.

The prioritization of selection criteria for compounds in this study has to be better explained. First, out of 249 compounds the authors selected 35 compounds based on their effect on iron induced neurotoxicity. Out of these 35 they selected 23 compounds based on the effect on rotenone induced neurotoxicity. Out of these 23 compounds, the authors chose 1 compound (or a class of 3

compounds) based on its effect on T cell proliferation. So implicitly the authors presume that in the pathomechanism of chronic MS, iron accumulation is more important than reactive oxygen, which again is more important than adaptive immune responses. Although individually all mechanisms do participate in chronic MS, I do not think that this implicit prioritization of mechanisms is well founded. The main outcome of the study would most likely have been different if they had inverted their order of screening selection.

Related to this aspect are the following questions: Can the priority and ordering of criteria be operationalized and can the quality of prioritization be tested? Could the authors calculate a merged predicted efficacy score? In other words would a compound with 120% efficacy in one assay and 70% efficacy in two other assays be more or less effective than a compound with 90% efficacy in all three assays?

Minor points

Fig. 8 e lacks an untreated/naïve control.

The authors should comment on the fact that preventive Clomipramine treatment fully abolishes clinical EAE (Fig. 8) while histological lesions do develop (Fig. 9). What is the presumed mechanism?

Figure 1B-d make the reader believe that Indapamide will be studied in greater detail later in the study. Why this representative drug?

I do not agree with the statement that chronic MS differs from RRMS mostly with 'respect to magnitude' of response. In my opinion they are mediated by different mechanisms.

Reference 19 cited in line 208 is inappropriate. It shows the presence of circulating TFH cells in MS, which may or may not be causative for inducing B cell follicles in the meninges. But other papers e.g. Magliozzi et al. 2007 demonstrate the actual presence of B cell follicles in MS.

A novel compound fused from the antidepressant imipramine and the anti-malaria drug quinacrine (named quinpramine) is efficacious in EAE (Singh et al. 2009 Exp Neurol) and this line of evidence should be discussed.

Reviewer #2 (Remarks to the Author):

This is a very interesting paper addressing an exciting and comprehensive screen of much needed compounds for the progressive form of Multiple Sclerosis. This is an area of current therapeutic challenge and intense investigation.

There are several strengths in the manuscript:

1. the use of human neurons and human B cells
2. the integration of screening methods for neuroprotection and immunomodulation
3. the identification of compounds that are already being used for other pathology, thereby bearing the promise of potential repurposing

Some weaknesses include:

1. the reliance on a single model of EAE (a relatively acute model based on MOG-peptide immunization of C57-Bl6 mice).
2. the lack of clinical endpoints supporting a sustained neuroprotective effect in mice treated with clomipramine. If the treated mice over a relatively short period of time reach the same level of disability as the untreated controls, how do the authors envision this drug to be translated to humans?

Points to be addressed:

1. test other drugs since Clomipramine has an overall minor effect in vitro and therefore is not surprising mled to very minor effects in vivo. Overall Liothyronine and Atenolol or Carvedilol seem to have a stronger neuroprotective effect against iron neurotoxicity and mitochondrial damage. The lack of effect on B cells or immune cells could even be beneficial as therapies targeting B cells are available and could be combined
2. test the effect of the successful lead compound in at least an additional of chronic progressive demyelination (MOG immunization in NOD mice as model of secondary progressive MS; chronic cuprizone as model of axonal damage in the absence of BBB permeability; etc.) or in other mouse models of neurodegeneration
3. since treatment would start after neurons have been already exposed to some sort of "insult", it is important to consider including the effect of post-treatment or co-treatment rather than pre-treatment

Minor comments:

1. please clarify and define the word " control". Distinguish between untreated control or time 0 of given treatment (very confusing especially in Figure legend 2)
2. please be consistent with the order of the drugs so that compounds tested for one feature in one figure can be easily compared with the effects tested in other figures
3. please clarify why human neurons and B cells were used but T cells were isolated from mice
4. please clarify why "control " neurons are dying over time in Fig 2
5. please clarify why Sidak's test was chosen in Fig. 3
4. please clarify why

Reviewer #3 (Remarks to the Author):

This was an interesting study in which a series of screening procedures identified CMI as a potential treatment for MS from among the NINDS Compound Library.

CMI was selected from 1040 compounds within the NINDS library. This is a large number of compounds, but it is a relatively small library relative to the libraries maintained by large pharmaceutical industry companies. This made me wonder whether the limited novelty of the classes of drugs identified as active were a by-product of the limited library available to the investigators.

Each screening step was selected on the basis of a putative contribution to MS-related neurotoxicity (Fe toxicity, rotenone toxicity, free radical injury, T cell proliferation, and B cell proliferation). My read of recent reviews in this area (Beiske et al. MS Int 2015, etc.) suggests that some of these toxicities are well described in MS (Fe for example), but it not yet clear that they all play a major role in disease progression. Further, it is not entirely clear what degree of activity in each of these assays is needed to affect clinical outcomes. Thus, it is not clear how the results of the various assays should be weighted when considering the candidates emerging from their screens.

In these assays, TCAs and antipsychotics were generally effective. CMI was selected for further study not because it had unique properties, but because it had a generally good profile. In their EAE, CMI produced promising results as well.

I was struck that the elaborate selection process in this paper identified TCAs as a promising class of medications for the treatment of MS. These medications have been widely prescribed for pain and mood symptoms in MS. However, I could not find any reports of disease-modifying effects of TCAs for MS. Is that because TCAs are ineffective for MS or just that no one looked carefully?

With regards to the former possibility, the transient activity of CMI in the EAE model might not translate to meaningful clinical benefit. It seems unlikely that we have missed a "disruptive

advance" with respect to the clinical utility of TCAs. For example, TCAs have some utility for pain and for depression in MS, but even for these outcomes there are high rates of partial and non-response. Nonetheless, it is possible that some more subtle benefits of TCAs or benefits for subpopulations of MS patients have been overlooked.

Revision NCOMMS-16-28655

We thank the editor and the reviewers for their excellent and constructive comments, which clearly helped to improve the quality of this manuscript. We have performed additional experiments as detailed below, thereby addressing the issues raised by the reviewers. We are pleased to provide the following point-to-point reply.

In response to Reviewer #1:

1. *“In this experimental study, the authors established a high-throughput screening assay to identify novel potential pharmaceutical compounds for the treatment of progressive multiple sclerosis (MS). They based their screening on measures of neurotoxicity, mitochondrial damage, reactive oxygen species, and T cell proliferation. Subsequently, they used this assay to screen a large drug collection and identify tricyclic anti-depressants as promising new drug candidates. From this class of drugs the authors chose clomipramine to demonstrate the efficacy in an animal model of MS.*

Overall this is an excellently written and very well performed experimental study that establishes a novel screening approach and identifies a novel class of compounds. On the one hand, the study thus creates the necessary tools to screen and validate novel compounds for the future treatment of a debilitating human disorder. On the other hand, the study sets the stage for testing an entire class of drugs in progressive MS. A speculative immediate implication of this study could be to preferentially use tricyclic rather than other types of antidepressants for the treatment of depression in progressive MS patients. Overall, this is a highly important, exciting and innovative study and its findings clearly deserve publication and further investigation.”

We thank the Reviewer for these laudatory comments.

2. *“Major points: What should be better explained is, how the prioritization of screening approaches is to be understood, and how the innovative and original screening can be better linked to a good in vivo model for plausibilization of the mechanisms linked.”*

We have now explained in the Discussion (page 13) that there was no prioritization of screening approaches, as the main pathophysiology of progressive MS is unknown, and that this was a limitation of the study. Please also see the rebuttal to point #5 below.

3. *“The authors emphasize the applicability of their screening approach to chronic MS. But then they*

use MOG35-55 peptide induced EAE in C57BL/6 mice and use a follow-up of 19 and 15 days respectively to illustrate the applicability of Clomipramine in MS. The use of this specific animal model in this context can be discussed critically. If at all, this EAE variant is a model of one acute MS relapse especially if followed-up for such brief periods of time. Axonal loss in MOG peptide induced EAE correlates closely with peak disease severity and this does not prove a neuroprotective effect of the compound. Following the animals for only 15 days (Fig. 8) has limited value given that the peak severity of EAE seems to be delayed rather than suppressed in Figure 7.

Various models have been developed that reproduce some aspects of chronic MS e.g. actively induced EAE in Biozzi ABH mice (as discussed in line 338), in NOD mice (used e.g. in Mayo et al. 2015 Nat Med), in DA rats, in SJL mice or spontaneous models such as the 2D2xTH MOG TCR and BCR double transgenic mouse line. Overall, the second part of this study shows a clear drop in quality and does not support using Clomipramine in chronic MS at all. So, the authors are advised to use and implement at least one additional model, that better mimics aspects of „progressive“ nature of the disease”

We thank the reviewer for this insightful and excellent remark. As advised by the Reviewer, we have now performed additional experiments of clomipramine in chronic treatment paradigms. In the first experiment in chronic MOG-induced EAE in C57BL/6 mice (Fig. 10a), we started treatment at day 30 during the remission phase after mice had their first relapse. Here, we could not document a positive effect for clomipramine; we have postulated (page 11) that this may be related to the late treatment in mice that have already accumulated substantial injury from a prolonged EAE course. In the second experiment, we treated animals from the onset of clinical signs and found that clomipramine reduced the severity of the first relapse. During the remission phase, likely because the severity of disability was low, no apparent rescue by clomipramine was evident. However, when the second relapse occurred, clomipramine prevented this relapse (Fig. 10b). These results support a therapeutic role for clomipramine in chronic EAE.

We note that while the MOG-induced C57BL/6 EAE model is generally considered a monophasic disease, we find that by using a 15-point scoring scale (instead of the commonly used 5-point but less sensitive scale), a period of remission followed by further worsening can be detected. We have stated this on page 11.

To test the effects of clomipramine further in chronic EAE, we have used the Biozzi ABH mouse model that some have argued to be a reasonable model of progressive MS. Here, as shown in Figure 10c, treatment of mice with clomipramine when clinical signs have manifested reduced the subsequent progression of disability that was observed in EAE mice treated with vehicle.

We also tested the effects of clomipramine in the NOD model but mice did not succumb to an EAE phenotype. That experiment was thus aborted.

Overall, our new results (new Fig. 10) support the utility of clomipramine in reducing disability in chronic paradigms: the chronic MOG-induced and Biozzi ABH models. While making this point, we have also noted in the Discussion (page 13) that there are currently no perfect models for the progressive forms of MS.

4. *“Part of the authors’ selection criteria for compounds was oral bio-availability (line 111). In a real world scenario this would eliminate for example Ocrelizumab from a list of potential compounds and there is no good reason to believe that an effective treatment for chronic MS would HAVE to be oral. This narrows the list of compounds unnecessarily.”*

We agree with the reviewer that an effective treatment for chronic MS need not have to be an oral formulation. We have now made this point on page 7, and emphasized that the choice for an oral medication is simply for its ease of use.

5. *“The prioritization of selection criteria for compounds in this study has to be better explained. First, out of 249 compounds the authors selected 35 compounds based on their effect on iron induced neurotoxicity. Out of these 35 they selected 23 compounds based on the effect on rotenone induced neurotoxicity. Out of these 23 compounds, the authors chose 1 compound (or a class of 3 compounds) based on its effect on T cell proliferation. So implicitly the authors presume that in the pathomechanism of chronic MS, iron accumulation is more important than reactive oxygen, which again is more important than adaptive immune responses. Although individually all mechanisms do participate in chronic MS, I do not think that this implicit prioritization of mechanisms is well founded. The main outcome of the study would most likely have been different if they had inverted their order of screening selection.*

Related to this aspect are the following questions: Can the priority and ordering of criteria be operationalized and can the quality of prioritization be tested? Could the authors calculate a merged predicted efficacy score? In other words would a compound with 120% efficacy in one assay and 70% efficacy in two other assays be more or less effective than a compound with 90% efficacy in all three assays?”

Again, the Reviewer has made an excellent point. We now state clearly (on page 14) that the sequence of the screens was not a reflection of which pathophysiology was most important for progressive MS, but simply because the protection of neurons against death in culture was an easily observed (through microscopy of living neurons) and therefore obvious outcome. We also stated the cautionary note that had the screen been sequenced in a different manner, other lead candidates might have emerged. Nonetheless, through the sequence that we employed, we were pleased to uncover clomipramine that then showed effectiveness in mice.

We did not calculate a merged predicted efficacy score, albeit a great idea, as this would presuppose that all the outcomes have specific degrees of importance in progressive MS, which is not currently

known.

6. *“Minor points, Fig. 8 e lacks an untreated/naïve control.”*

The vehicle control group (ie no clomipramine) has now been added to Figure 8e, and it shows no detectable clomipramine or DMCL levels.

7. *“The authors should comment on the fact that preventive Clomipramine treatment fully abolishes clinical EAE (Fig. 8) while histological lesions do develop (Fig. 9). What is the presumed mechanism?”*

Thank you for this astute observation. We now describe in the Results (page 11) that a histological score of 1.7 (a few inflammatory cells in the meninges of clomipramine-treated mice, and which have not infiltrated into the spinal cord parenchyma) is inadequate to produce clinical manifestations.

8. *“Figure 1B-d make the reader believe that Indapamide will be studied in greater detail later in the study. Why this representative drug?”*

Indeed, indapamide was very effective in protective against iron neurotoxicity (Fig. 1,2) or as a free radical scavenger (Fig. 4). Thus, we highlighted this drug initially as one example of potential neuroprotective agents. However, we found subsequently (Fig. 5) that it did not affect the proliferation of T cells while clomipramine (which was also neuroprotective) did. This, we focused on clomipramine in subsequent experiments. We have now explained on page 9 that “As indapamide did not reduce T cell proliferation, we did pursue it further in the T cell-prominent disease, EAE”.

9. *“I do not agree with the statement that chronic MS differs from RRMS mostly with ‘respect to magnitude’ of response. In my opinion they are mediated by different mechanisms”.*

We appreciate that there are differing considerations of how the pathology of RRMS and progressive MS might differ. Hence, we deleted this part of the sentence.

10. *“Reference 19 cited in line 208 is inappropriate. It shows the presence of circulating TFH cells in MS, which may or may not be causative for inducing B cell follicles in the meninges. But other papers e.g. Magliozzi et al. 2007 demonstrate the actual presence of B cell follicles in MS.”*

Thank you for pointing this out. The reference is changed to Magliozzi et al. 2007.

11. *“A novel compound fused from the antidepressant imipramine and the anti-malaria drug quinacrine (named quinpramine) is efficacious in EAE (Singh et al. 2009 Exp Neurol) and this line of evidence should be discussed. “*

This reference (#32) was added to the discussion with this sentence: “A novel compound recently developed, quinpramine, which is a fusion of imipramine and the anti-malarial quinacrine, decreased the number of inflammatory CNS lesions, antigen-specific T-cell proliferation and pro-inflammatory cytokines in EAE”.

In response to Reviewer #2:

1. *“This is a very interesting paper addressing an exciting and comprehensive screen of much needed compounds for the progressive form of Multiple Sclerosis. This is an area of current therapeutic challenge and intense investigation. There are several strengths in the manuscript: 1. the use of human neurons and human B cells, 2. the integration of screening methods for neuroprotection and immunomodulation, 3. the identification of compounds that are already being used for other pathology, thereby bearing the promise of potential repurposing. Some weaknesses include: 1. the reliance on a single model of EAE (a relatively acute model based on MOG-peptide immunization of C57-B16 mice). 2. the lack of clinical endpoints supporting a sustained neuroprotective effect in mice treated with clomipramine. If the treated mice over a relatively short period of time reach the same level of disability as the untreated controls, how do the authors envision this drug to be translated to humans? Points to be addressed: 1. test other drugs since Clomipramine has an overall minor effect in vitro and therefore is not surprising led to very minor effects in vivo. Overall Liothyronine and Atenolol or Carvedilol seem to have a stronger neuroprotective effect against iron neurotoxicity and mitochondrial damage. The lack of effect on B cells or immune cells could even be beneficial as therapies targeting B cells are available and could be combined.”*

We thank the Reviewer for commenting that this is a very interesting paper, and for pointing out the strengths and weaknesses. With regards to the weaknesses of the reliance on a single model or the lack of demonstration of a sustained effect, we have now rectified this by using the MOG-EAE model in a more chronic setting, and by using the Biozzi ABH model (new Fig. 10) (see also our rebuttal to Reviewer 1, point #3).

With regards to testing other drugs, we wish first to emphasize that clomipramine showed a strong effect in the majority of the in vitro assays. The neuroprotective effect was strong (Figure 2a, 107.3% of control neurons, meaning complete rescue against the toxicity of iron), the anti-oxidative effect was

stronger than that of the positive control anti-oxidant gallic acid (Figure 4c, HORAC-GAE 2.1, or a doubling effect) and the T-cell proliferation was reduced by 68.2% (Figure 5). The inhibition of B cell proliferation was complete at 5 μ M (Figure 6g).

With regards to Liothyronine and Atenolol or Carvedilol, these do not penetrate the CNS (probability of 68% for all three) as well as clomipramine (97.9% chance for entering the CNS according to drugbank.ca). They also do not affect T-cell proliferation (Fig. 5). This is now discussed on page 15.

We agree that it might be interesting to combine the medication with e. g. B-cell depleting agents. However, clomipramine itself has B-cell inhibitory activity (Fig. 6g,h).

2. *“test the effect of the successful lead compound in at least an additional of chronic progressive demyelination (MOG immunization in NOD mice as model of secondary progressive MS; chronic cuprizone as model of axonal damage in the absence of BBB permeability; etc.) or in other mouse models of neurodegeneration.”*

As stated above, we performed additional experiments with chronic MOG-EAE and in the Biozzi ABH model. We did test clomipramine in the NOD model but mice did not succumb to an EAE phenotype so the experiment had to be aborted.

3. *“since treatment would start after neurons have been already exposed to some sort of "insult", it is important to consider including the effect of post-treatment or co-treatment rather than pre-treatment”*

In tissue culture, the toxicity of iron to neurons begins immediately. It has been our experience that pretreatment with test protective agents is thus necessary. In the methods (page 18), we now state this point, and also included the justification that with the continuous insult that occurs in MS, a pretreatment paradigm simulates the protection against the next injury. For the C56BL/6 mice and Biozzi ABH mouse experiments, treatment is initiated days after the immune system has been activated, such as at the onset of clinical signs of disease.

4. *“Minor comments: 1. please clarify and define the word " control". Distinguish between untreated control or time 0 of given treatment (very confusing especially in Figure legend 2).*

We apologize for the confusion. Where indicated, we now clarify this in the respective figure legends.

5. *“2, please be consistent with the order of the drugs so that compounds tested for one feature in one figure can be easily compared with the effects tested in other figures”.*

Thank you. We have changed the order of doxepin in figures 3 and 5 (before imipramine); the rest of the medications is presented in the same order.

6. *“3. please clarify why human neurons and B cells were used but T cells were isolated from mice*

This was due to feasibility and convenience. Human neurons could be cultured in large numbers and thus could be used in multiple experiments. To perform a high throughput screen with human T cells, we would have to rely on frequent bleeds of human volunteers; thus, we resorted to murine T-cells due to the convenience of obtaining mice to extract cells at any time. For human B cells that were used to address the impact of clomipramine, this needed small number of cells and thus we returned to the human source.

7. *“4. please clarify why "control " neurons are dying over time in Fig 2”*

The live cell imaging ImageXpress system offers the possibility to investigate cells under relatively stable conditions (5% CO₂, 37°C). Those conditions are, however, far from optimal and do not reach the conditions in an incubator (no humidity in the ImageXpress, temperature fluctuation, CO₂ pressure varies stronger). Thus, also control neurons die over time.

8. *“5. please clarify why Sidak's test was chosen in Fig. 3”*

Sidak's test was chosen due to power, but we have changed the analysis to Bonferroni, which did not change the level of significance.

In response to Reviewer #3:

1. *“This was an interesting study in which a series of screening procedures identified CMI as a potential treatment for MS from among the NINDS Compound Library.*

CMI was selected from 1040 compounds within the NINDS library. This is a large number of compounds, but it is a relatively small library relative to the libraries maintained by large pharmaceutical industry companies. This made me wonder whether the limited novelty of the classes of drugs identified as active were a by-product of the limited library available to the investigators.”

We agree with this reviewer that the number of compounds which could be tested is not exhaustive and does not reach the high-throughput screenings which can be performed by big entities. However, it was not our intention to find new and novel compounds which might be neuroprotective. Rather, we wanted to focus on generics as their potential to advance faster into clinical studies is much higher, given that there would be knowledge of their spectrum of side effects in human use. The NINDS library was advertised as a library of mostly generic medications, which is why we proceeded to use that. This is now discussed on page 14. While we agree that the current datasets are limited, we are nonetheless excited that several candidate hits did emerge which can now be potentially taken into clinical trials.

2. “Each screening step was selected on the basis of a putative contribution to MS-related neurotoxicity (Fe toxicity, rotenone toxicity, free radical injury, T cell proliferation, and B cell proliferation). My read of recent reviews in this area (Beiske et al. MS Int 2015, etc.) suggests that some of these toxicities are well described in MS (Fe for example), but it not yet clear that they all play a major role in disease progression. Further, it is not entirely clear what degree of activity in each of these assays is needed to affect clinical outcomes. Thus, it is not clear how the results of the various assays should be weighted when considering the candidates emerging from their screens.”

We thank the reviewer for this excellent point. Indeed, it remains a conundrum to which degree different pathogenic mechanisms drive progression. Therefore, this question cannot be answered properly. However, we believe that the combination of different assays with features seen in progressive MS helps to define medications which might be efficacious in progression. Therefore, we stated in the discussion “We hope that the results of the current study will lead to a clinical trial in the ultimate test subjects: patients with progressive multiple sclerosis.” (page 13). Please also see our rebuttal (point #5) to Reviewer #1 where we discuss that it would be difficult to weigh the relative importance of the different screens since we do not know which pathologic feature would be most important in driving progression in MS.

3. “In these assays, TCAs and antipsychotics were generally effective. CMI was selected for further study not because it had unique properties, but because it had a generally good profile. In their EAE, CMI produced promising results as well. I was struck that the elaborate selection process in this paper identified TCAs as a promising class of medications for the treatment of MS. These medications have been widely prescribed for pain and mood symptoms in MS. However, I could not find any reports of disease-modifying effects of TCAs for MS. Is that because TCAs are ineffective for MS or just that no one looked carefully? Discussion point for Simon, Marcus and Luanne to address

With regards to the former possibility, the transient activity of CMI in the EAE model might not

translate to meaningful clinical benefit. It seems unlikely that we have missed a "disruptive advance" with respect to the clinical utility of TCAs. For example, TCAs have some utility for pain and for depression in MS, but even for these outcomes there are high rates of partial and non-response. Nonetheless, it is possible that some more subtle benefits of TCAs or benefits for subpopulations of MS patients have been overlooked."

This is an excellent point raised by the Reviewer. One explanation might be that progression occurs relatively slow over a longer period of time and clinical changes might only be subtle. Hence, it is difficult for clinicians to measure therapeutic effects. Even in big clinical trials differences are small, as just recently shown in the ORATORIO trial. Here, the positive effect of ocrelizumab led only to roughly 20% less patients with reduced 24-month disability progression. Hence, distinct from RRMS where therapeutic effects are relatively easy to measure either clinically or radiologically, it might be possible that effects have been missed until now.

REVIEWERS' COMMENTS:

Reviewer #1 (Remarks to the Author):

The authors now present a revised version of their manuscript 'Systematic screening of generic drugs for progressive multiple sclerosis: Clomipramine as a promising therapeutic' which has been submitted to Nature Communications.

The authors now provide an additional figure 10 to convincingly demonstrate that Clomipramine is not only effective in short-term MOG peptide-induced EAE, but also ameliorates chronic EAE models. This was the most important previous concerns with the study. All other comments have also been adequately addressed. Overall, the authors have thus extensively revised their manuscript and thoroughly addressed my concerns. I recommend accepting the manuscript in its present form.

I only have one mandatory minor correction before publication:

The asterisks in Figure 10 panels B and C seem to indicate that the average score between treatment and vehicle is significantly different when averaged between days 42 and 50 in panel B and between day 14 and day 30 in panel C. If this is indeed the case, the statistical calculation is inadequate. A test for significance needs to be performed for each individual day as the authors have done in Figure 7A and 8A. Also, the description of the statistical approach in the legend of Figure 10 needs to be improved.

Reviewer #2 (Remarks to the Author):

This is a thorough revision of a manuscript which presents very exciting findings. The authors provide a substantial number of new experiments that - in my opinion- have addressed all the previous concerns and tilted the weakness to strength balance significance towards the latter one. I believe this is an important paper definitely deserving further consideration

Reviewer #3 (Remarks to the Author):

The authors have reasonably addressed my concerns.

Authors point-by-point rebuttal:

In response to Reviewer #1:

1. *“The authors now present a revised version of their manuscript ‘Systematic screening of generic drugs for progressive multiple sclerosis: Clomipramine as a promising therapeutic’ which has been submitted to Nature Communications. The authors now provide an additional figure 10 to convincingly demonstrate that Clomipramine is not only effective in short-term MOG peptide-induced EAE, but also ameliorates chronic EAE models. This was the most important previous concerns with the study. All other comments have also been adequately addressed. Overall, the authors have thus extensively revised their manuscript and thoroughly addressed my concerns. I recommend accepting the manuscript in its present form.”*

We thank the referee for this kind comment.

2. *“I only have one mandatory minor correction before publication: The asterisks in Figure 10 panels B and C seem to indicate that the average score between treatment and vehicle is significantly different when averaged between days 42 and 50 in panel B and between day 14 and day 30 in panel C. If this is indeed the case, the statistical calculation is inadequate. A test for significance needs to be performed for each individual day as the authors have done in Figure 7A and 8A. Also, the description of the statistical approach in the legend of Figure 10 needs to be improved.”*

We thank the reviewer for remarking on this important point and we apologize for the previous lack of clarity of the statistical approach in the legend to Figure 10. We now state that for panel B, an initial two-way ANOVA with Sidak’s multiple-comparisons test (as done for Figure 7A and 8A as noted by the referee) of the experiment from day 13 to 50 was not statistically significant, since vehicle-treated mice spontaneously remitted to a very low disease score between days 25 and 42, so that differences with the clomipramine-treated group could not be detected. Hence, we analyzed the results of the acute and chronic relapse phases outside of the period of remission, using Mann-Whitney t-test. This matches what we had done for Figure 7B, where the disease activity of 2 groups over a specified period of time was contrasted. Besides now clarifying the statistical method and its rationale in the Figure legend, we have also mentioned in the Discussion (lines 366 to 372) this limitation of the statistical analysis of the chronic experiments, where a period of remission prevented the statistical significance across the entire experiment to manifest. We also noted in that paragraph that the differences of the acute and chronic relapse phases outside of the period of remission would be more meaningful for a drug’s utility in MS.

For Figure 10c, the model of Biozzi ABH chronic EAE is very variable in our hands. Day to day variability in disease score of mice within a group was high, so that when a two-way ANOVA with Sidak’s multiple-comparisons test was used, the results were not significant. Thus, we resorted to the use of the two-tailed Mann-Whitney t-test. These considerations and rationale are now noted in the figure legend.